# Learning Representations of Sets through Optimized Permutations

**Yan Zhang & Adam Prügel-Bennett & Jonathon Hare**
Department of Electronics and Computer Science
University of Southampton
`{yz5n12,apb,jsh2}@ecs.soton.ac.uk`

## Abstract

Representations of sets are challenging to learn because operations on sets should be permutation-invariant. To this end, we propose a *Permutation-Optimisation* module that learns how to permute a set end-to-end. The permuted set can be further processed to learn a permutation-invariant representation of that set, avoiding a bottleneck in traditional set models. We demonstrate our model's ability to learn permutations and set representations with either explicit or implicit supervision on four datasets, on which we achieve state-of-the-art results: number sorting, image mosaics, classification from image mosaics, and visual question answering.

## 1 Introduction

Consider a task where each input sample is a *set* of feature vectors with each feature vector describing an object in an image (for example: person, table, cat). Because there is no a priori ordering of these objects, it is important that the model is invariant to the order that the elements appear in the set. However, this puts restrictions on what can be learned efficiently. The typical approach is to compose elementwise operations with permutation-invariant reduction operations, such as summing (Zaheer et al., 2017) or taking the maximum (Qi et al., 2017) over the whole set. Since the reduction operator compresses a set of any size down to a single descriptor, this can be a significant bottleneck in what information about the set can be represented efficiently (Qi et al., 2017; Le & Duan, 2018; Murphy et al., 2019).

We take an alternative approach based on an idea explored in Vinyals et al. (2015a), where they find that some permutations of sets allow for easier learning on a task than others. They do this by ordering the set elements in some predetermined way and feeding the resulting sequence into a recurrent neural network. For instance, it makes sense that if the task is to output the top-n numbers from a set of numbers, it is useful if the input is already sorted in descending order before being fed into an RNN. This approach leverages the representational capabilities of traditional sequential models such as LSTMs, but requires some prior knowledge of what order might be useful.

Our idea is to learn such a permutation purely from data *without requiring a priori knowledge* (section 2). The key aspect is to turn a set into a sequence in a way that is both permutation-invariant, as well as differentiable so that it is learnable. Our main contribution is a Permutation-Optimisation (PO) module that satisfies these requirements: it optimises a permutation in the forward pass of a neural network using pairwise comparisons. By feeding the resulting sequence into a traditional model such as an LSTM, we can learn a flexible, permutation-invariant representation of the set while avoiding the bottleneck that a simple reduction operator would introduce. Techniques used in our model may also be applicable to other set problems where permutation-invariance is desired, building on the literature of approaches to dealing with permutation-invariance (section 3).

In four different experiments, we show improvements over existing methods (section 4). The former two tasks measure the ability to learn a particular permutation as target: number sorting and image mosaics. We achieve state-of-the-art performance with our model, which shows that our method is suitable for representing permutations in general. The latter two tasks test whether a model can learn to solve a task that requires it to come up with a suitable permutation implicitly: classification from image mosaics and visual question answering. We provide no supervision of what the permutation should be; the model has to learn by itself what permutation is most useful for the task at hand.

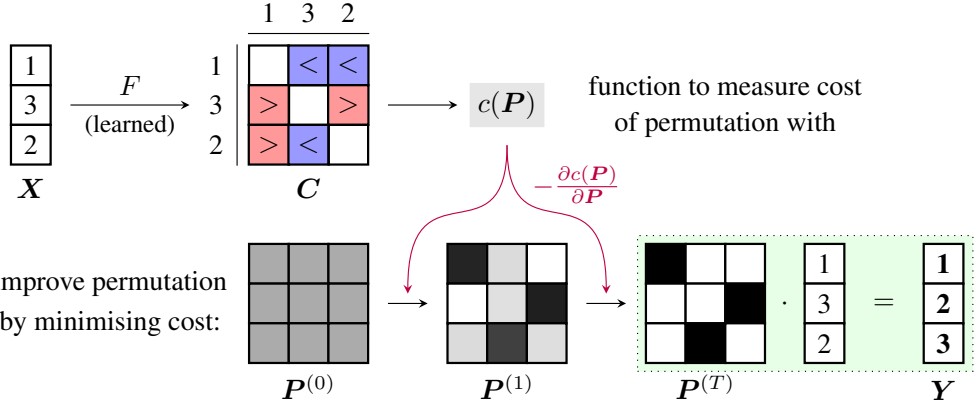

Figure 1: Overview of Permutation-Optimisation module. In the ordering cost $\boldsymbol{C}$, elements of $\boldsymbol{X}$ are compared to each other (blue represents a negative value, red represents a positive value). Gradients are applied to unnormalised permutations $\widetilde{\boldsymbol{P}}^{(t)}$, which are normalised to proper permutations $\boldsymbol{P}^{(t)}$.

Here, our model also beats the existing models and we improve the performance of a state-of-the-art model in VQA with it. This shows that our PO module is able to learn good permutation-invariant representations of sets using our approach.

## 2 PERMUTATION-OPTIMISATION MODULE

We will now describe a differentiable, and thus learnable model to turn an *unordered* set $\{\boldsymbol{x}_i\}_N$ with feature vectors as elements into an *ordered* sequence of these feature vectors. An overview of the algorithm is shown in Figure 1 and pseudo-code is available in Appendix A. The input set is represented as a matrix $\boldsymbol{X} = [\boldsymbol{x}_1, \ldots, \boldsymbol{x}_N]^T$ with the feature vectors $\boldsymbol{x}_i$ as rows in some arbitrary order. In the algorithm, it is important to not rely on the arbitrary order so that $\boldsymbol{X}$ is correctly treated as a set. The goal is then to learn a permutation matrix $\boldsymbol{P}$ such that when permuting the rows of the input through $\boldsymbol{Y} = \boldsymbol{P}\boldsymbol{X}$, the output is ordered correctly according to the task at hand. When an entry $P_{ik}$ takes the value 1, it can be understood as assigning the $i$th element to the $k$th position in the output.

Our main idea is to first relate pairs of elements through an *ordering cost*, parametrised with a neural network. This pairwise cost tells us whether an element $i$ should preferably be placed before or after element $j$ in the output sequence. Using this, we can define a *total cost* that measures how good a given permutation is (subsection 2.1). The second idea is to optimise this total cost in each forward pass of the module (subsection 2.2). By minimising the total cost of a permutation, we improve the quality of a permutation with respect to the current ordering costs. Crucially, the ordering cost function – and thus also the total cost function – is learned. In doing so, the module is able to learn how to generate a permutation as is desired.

In order for this to work, it is important that the optimisation process itself is differentiable so that the ordering cost is learnable. Because permutations are inherently discrete objects, a continuous relaxation of permutations is necessary. For optimisation, we perform gradient descent on the total cost for a fixed number of steps and unroll the iteration, similar to how recurrent neural networks are unrolled to perform backpropagation-through-time. Because the *inner gradient* (total cost differentiated with respect to permutation) is itself differentiable with respect to the ordering cost, the whole model is kept differentiable and we can train it with a standard supervised learning loss.

Note that as long as the ordering cost is computed appropriately (subsection 2.3), all operations used turn out to be permutation-invariant. Thus, we have a model that respects the symmetries of sets while producing an output without those symmetries: a sequence. This can be naturally extended to outputs where the target is not a sequence, but grids and lattices (subsection 2.4).

## 2.1 TOTAL COST FUNCTION

The total cost function measures the quality of a given permutation and should be lower for better permutations. Because this is the function that will be optimised, it is important to understand what it expresses precisely.

The main ingredient for the total cost of a permutation is the pairwise ordering cost (details in subsection 2.3). By computing it for all pairs, we obtain a cost matrix $C$ where the entry $C_{ij}$ represents the ordering cost between $i$ and $j$: the cost of placing element $i$ anywhere before $j$ in the output sequence. An important constraint that we put on $C$ is that $C_{ij} = -C_{ji}$. In other words, if one ordering of $i$ and $j$ is "good" (negative cost), then the opposite ordering obtained by swapping them is "bad" (positive cost). Additionally, this constraint means that $C_{ii} = 0$. This makes sure that two very similar feature vectors in the input will be similarly ordered in the output because their pairwise cost goes to $0$.

In this paper we use a straightforward definition of the total cost function: a sum of the ordering costs over all pairs of elements $i$ and $j$. When considering the pair $i$ and $j$, if the permutation maps $i$ to be before $j$ in the output sequence, this cost is simply $C_{ij}$. Vice versa, if the permutation maps $i$ to be after $j$ in the output sequence, the cost has to be flipped to $C_{ji}$. To express this idea, we define the total cost $c \colon \mathbb{R}^{N \times N} \mapsto \mathbb{R}$ of a permutation $P$ as:

$$c(\boldsymbol{P}) = \sum_{ij} C_{ij} \sum_{k} P_{ik} \left( \sum_{k'>k} P_{jk'} - \sum_{k'<k} P_{jk'} \right) \tag{1}$$

This can be understood as follows: If the permutation assigns element $i$ to position $u$ (so $P_{iu} = 1$) and element $j$ to position $v$ (so $P_{jv} = 1$), the sums over $k$ and $k'$ simplify to $1$ when $v > u$ and $-1$ when $v < u$; permutation matrices are binary and only have one $1$ in any row and column, so all other terms in the sums are $0$. That means that the term for each $i$ and $j$ becomes $C_{ij}$ when $v > u$ and $-C_{ij} = C_{ji}$ when $v < u$, which matches what we described previously.

## 2.2 OPTIMISATION PROBLEM

Now that we can compute the total cost of a permutation, we want to optimise this cost with respect to a permutation. After including the constraints to enforce that $P$ is a valid permutation matrix, we obtain the following optimisation problem:

$$\begin{aligned}
\underset{\boldsymbol{P}}{\text{minimize}} \quad & c(\boldsymbol{P}) \\
\text{subject to} \quad & \forall i, k \colon P_{ik} \in \{0, 1\}, \\
& \forall i \colon \sum_{k} P_{ik} = 1, \sum_{k} P_{ki} = 1
\end{aligned} \tag{2}$$

Optimisation over $P$ directly is difficult due to the discrete and combinatorial nature of permutations. To make optimisation feasible, a common relaxation is to replace the constraint that $P_{ik} \in \{0, 1\}$ with $P_{ik} \in [0, 1]$ (Fogel et al., 2013). With this change, the feasible set for $P$ expands to the set of doubly-stochastic matrices, known as the Birkhoff or assignment polytope. Rather than hard permutations, we now have soft assignments of elements to positions, analogous to the latent assignments when fitting a mixture of Gaussians model using Expectation-Maximisation.

Note that we do not need to change our total cost function after this relaxation. Instead of discretely flipping the sign of $C_{ij}$ depending on whether element $i$ comes before $j$ or not, the sums over $k$ and $k'$ give us a weight for each $C_{ij}$ that is based on how strongly $i$ and $j$ are assigned to positions. This weight is positive when $i$ is on average assigned to earlier positions than $j$ and negative vice versa.

In order to perform optimisation of the cost under our constraints, we reparametrise $P$ with the Sinkhorn operator $S$ from Adams & Zemel (2011) (defined in Appendix B) so that the constraints are always satisfied. We found this to lead to better solutions than projected gradient descent in initial

experiments. After first exponentiating all entries of a matrix, $S$ repeatedly normalises all rows, then all columns of the matrix to sum to 1, which converges to a doubly-stochastic matrix in the limit.

$$\boldsymbol{P} = S(\widetilde{\boldsymbol{P}}) \tag{3}$$

This ensures that $\boldsymbol{P}$ is always approximately a doubly-stochastic matrix. $\widetilde{\boldsymbol{P}}$ can be thought of as the unnormalised permutation while $\boldsymbol{P}$ is the normalised permutation. By changing our optimisation to minimise $\widetilde{\boldsymbol{P}}$ instead of $\boldsymbol{P}$ directly, all constraints are always satisfied and we can simplify the optimisation problem to $\min_{\widetilde{\boldsymbol{P}}} c(\boldsymbol{P})$ without any constraints.

It is now straightforward to optimise $\widetilde{\boldsymbol{P}}$ with standard gradient descent. First, we compute the gradient:

$$\frac{\partial c(\boldsymbol{P})}{\partial P_{pq}} = 2 \sum_j C_{pj} \left( \sum_{k'>q} P_{jk'} - \sum_{k'<q} P_{jk'} \right) \tag{4}$$

$$\frac{\partial c(\boldsymbol{P})}{\partial \widetilde{P}_{pq}} = \frac{\partial \boldsymbol{P}}{\partial \widetilde{P}_{pq}} \cdot \frac{\partial c(\boldsymbol{P})}{\partial \boldsymbol{P}} \tag{5}$$

From equation 4, it becomes clear that this gradient is itself differentiable with respect to the ordering cost $C_{ij}$, which allows it to be learned. In practice, both $\partial c(\boldsymbol{P})/\partial \widetilde{\boldsymbol{P}}$ as well as $\partial [\partial c(\boldsymbol{P})/\partial \widetilde{\boldsymbol{P}}]/\partial \boldsymbol{C}$ can be computed with automatic differentiation. However, some implementations of automatic differentiation require the computation of $c(\boldsymbol{P})$ which we do not use. In this case, implementing $\partial c(\boldsymbol{P})/\partial \widetilde{\boldsymbol{P}}$ explicitly can be more efficient. Also notice that if we define $B_{jq} = \sum_{k'>q} P_{jk'} - \sum_{k'<q} P_{jk'}$, equation 4 is just the matrix multiplication $\boldsymbol{CB}$ and is thus efficiently computable.

For optimisation, $\boldsymbol{P}$ has to be initialised in a permutation-invariant way to preserve permutation-invariance of the algorithm. In this paper, we consider a uniform initialisation so that all $P_{ik} = 1/N$ (**PO-U** model, left) and an initialisation that linearly assigns (Mena et al., 2018) each element to each position (**PO-LA** model, right).

$$\widetilde{P}_{ik}^{(0)} = 0 \quad \text{or} \quad \widetilde{P}_{ik}^{(0)} = \boldsymbol{w}_k \boldsymbol{x}_i \tag{6}$$

where $\boldsymbol{w}_k$ is a different weight vector for each position $k$. Then, we perform gradient descent for a fixed number of steps $T$. The iterative update using the gradient and a (learnable) step size $\eta$ converges to the optimised permutation $\boldsymbol{P}^{(T)}$:

$$\widetilde{\boldsymbol{P}}^{(t+1)} = \widetilde{\boldsymbol{P}}^{(t)} - \eta \frac{\partial c(\boldsymbol{P}^{(t)})}{\partial \boldsymbol{P}^{(t)}} \tag{7}$$

One peculiarity of this is that we update $\widetilde{\boldsymbol{P}}$ with the gradient of the normalised permutation $\boldsymbol{P}$, not of the unnormalised permutation $\widetilde{\boldsymbol{P}}$ as normal. In other words, we do gradient descent on $\widetilde{\boldsymbol{P}}$ but in equation 5 we set $\partial P_{uv}/\partial \widetilde{P}_{pq} = 1$ when $u = p, v = q$, and 0 everywhere else. We found that this results in significantly better permutations experimentally; we believe that this is because $\partial \boldsymbol{P}/\partial \widetilde{\boldsymbol{P}}$ vanishes too quickly from the Sinkhorn normalisation, which biases $\boldsymbol{P}$ away from good permutation matrices wherein all entries are close to 0 and 1 (Appendix D).

The runtime of this algorithm is dominated by the computation of gradients of $c(\boldsymbol{P})$, which involves a matrix multiplication of two $N \times N$ matrices. In total, the time complexity of this algorithm is $T$ times the complexity of this matrix multiplication, which is $\Theta(N^3)$ in practice. We found that typically, small values for $T$ such as 4 are enough to get good permutations.

## 2.3 ORDERING COST FUNCTION

The ordering cost $C_{ij}$ is used in the total cost and tells us what the pairwise cost for placing $i$ before $j$ should be. The key property to enforce is that the function $F$ that produces the entries of $\boldsymbol{C}$ is anti-symmetric ($F(\boldsymbol{x}_i, \boldsymbol{x}_j) = -F(\boldsymbol{x}_j, \boldsymbol{x}_i)$). A simple way to achieve this is to define $F$ as:

$$F(\boldsymbol{x}_i, \boldsymbol{x}_j) = f(\boldsymbol{x}_i, \boldsymbol{x}_j) - f(\boldsymbol{x}_j, \boldsymbol{x}_i) \qquad (8)$$

We can then use a small neural network for $f$ to obtain a learnable $F$ that is always anti-symmetric.

Lastly, $\boldsymbol{C}$ is normalised to have unit Frobenius norm. This results in simply scaling the total cost obtained, but it also decouples the scale of the outputs of $F$ from the step size parameter $\eta$ to make optimisation more stable at inference time. $\boldsymbol{C}$ is then defined as:

$$\widetilde{C}_{ij} = F(\boldsymbol{x}_i, \boldsymbol{x}_j) \qquad (9)$$
$$C_{ij} = \widetilde{C}_{ij}/\|\widetilde{\boldsymbol{C}}\|_F \qquad (10)$$

## 2.4 EXTENDING PERMUTATIONS TO LATTICES

In some tasks, it may be natural to permute the set into a lattice structure instead of a sequence. For example, if it is known that the set contains parts of an image, it makes sense to arrange these parts back to an image by using a regular grid. We can straightforwardly adapt our model to this by considering each row and column of the target grid as an individual permutation problem. The total cost of an assignment to a grid is the sum of the total costs over all individual rows and columns of the grid. The gradient of this new cost is then the sum of the gradients of these individual problems. This results in a model that considers both row-wise and column-wise pairwise relations when permuting a set of inputs into a grid structure, and more generally, into a lattice structure.

## 3 RELATED WORK

The most relevant work to ours is the inspiring study by Mena et al. (2018), where they discuss the reparametrisation that we use and propose a model that can also learn permutations implicitly in principle. Their model uses a simple elementwise linear map from each of the $N$ elements of the set to the $N$ positions, normalised by the Sinkhorn operator. This can be understood as classifying each element individually into one of the $N$ classes corresponding to positions, then normalising the predictions so that each class only occurs once within this set. However, processing the elements individually means that their model does not take relations between elements into account properly; elements are placed in absolute positions, not relative to other elements. Our model differs from theirs by considering pairwise relations when creating the permutation. By basing the cost function on pairwise comparisons, it is able to order elements such that local relations in the output are taken into account. We believe that this is important for learning from permutations implicitly, because networks such as CNNs and RNNs rely on local ordering more than absolute positioning of elements. It also allows our model to process variable-sized sets, which their model is not able to do.

Our work is closely related to the set function literature, where the main constraint is invariance to ordering of the set. While it is always possible to simply train using as many permutations of a set as possible, using a model that is naturally permutation-invariant increases learning and generalisation capabilities through the correct inductive bias in the model. There are some similarities with relation networks (Santoro et al., 2017) in considering all pairwise relations between elements as in our pairwise ordering function. However, they sum over all non-linearly transformed pairs, which can lead to the bottleneck we mention in section 1. Meanwhile, by using an RNN on the output of our model, our approach can encode a richer class of functions: it can still learn to simply sum the inputs, but it can also learn more complex functions where the learned order between elements is taken into account. The concurrent work in (Murphy et al., 2019) discusses various approximations of averaging the output of a neural network over all possible permutations, with our method falling under their categorisation of a learned canonical input ordering. Our model is also relevant to neural networks operating on graphs such as graph-convolutional networks (Kipf & Welling, 2017). Typically, a set

function is applied to the set of neighbours for each node, with which the state of the node is updated. Our module combined with an RNN is thus an alternative set function to perform this state update with.

Noroozi & Favaro (2016) and Cruz et al. (2017) show that it is possible to use permutation learning for representation learning in a self-supervised setting. The model in Cruz et al. (2017) is very similar to Mena et al. (2018), including use of a Sinkhorn operator, but they perform significantly more processing on images with a large CNN (AlexNet) beforehand with the main goal of learning good representations for that CNN. We instead focus on using the permuted set itself for representation learning in a supervised setting.

We are not the first to explore the usefulness of using optimisation in the forward pass of a neural network (for example, Stoyanov et al. (2011); Domke (2012); Belanger et al. (2017)). However, we believe that we are the first to show the potential of optimisation for processing sets because – with an appropriate cost function – it is able to preserve permutation-invariance. In OptNet (Amos & Kolter, 2017), exact solutions to convex quadratic programs are found in a differentiable way through various techniques. Unfortunately, our quadratic program is non-convex (Appendix E), which makes finding an optimal solution possibly NP-hard (Pardalos & Vavasis, 1991). We thus fall back to the simpler approach of gradient descent on the reparametrised problem to obtain a non-optimal, but reasonable solution.

Note that our work differs from learning to rank approaches such as Burges et al. (2005) and Severyn & Moschitti (2015), as there the end goal is the permutation itself. This usually requires supervision on what the target permutation should be, producing a permutation with hard assignments at the end. We require our model to produce soft assignments so that it is easily differentiable, since the main goal is not the permutation itself, but processing it further to form a representation of the set being permuted. This means that other approaches that produce hard assignments such as Ptr-Net (Vinyals et al., 2015b) are also unsuitable for implicitly learning permutations, although using a variational approximation through Mena et al. (2018) to obtain a differentiable permutation with hard assignments is a promising direction to explore for the future. Due to the lack of differentiability, existing literature on solving minimum feedback arc set problems (Charon & Hudry, 2007) can not be easily used for set representation learning either.

## 4 EXPERIMENTS

Throughout the text, we will refer to our model with uniform assignment as PO-U, with linear assignment initialisation as PO-LA, and the model from Mena et al. (2018) as LinAssign. We perform a qualitative analysis of what comparisons are learned in Appendix C. Precise experimental details can be found in Appendix F and our implementation for all experiments is available at `https://github.com/Cyanogenoid/perm-optim` for full reproducibility. Some additional results including example image mosaic outputs can be found in Appendix G.

### 4.1 SORTING NUMBERS

We start with the toy task of turning a set of random unsorted numbers into a sorted list. For this problem, we train with fixed-size sets of numbers drawn uniformly from the interval $[0, 1]$ and evaluate on different intervals to determine generalisation ability (for example: $[0, 1], [0, 1000], [1000, 1001]$). We use the correctly ordered sequence as training target and minimise the mean squared error. Following Mena et al. (2018), during evaluation we use the Hungarian algorithm for solving a linear assignment problem with $-\boldsymbol{P}$ as the assignment costs. This is done to obtain a permutation with hard assignments from our soft permutation.

Our PO-U model is able to sort all sizes of sets that we tried – 5 to 1024 numbers – perfectly, including generalising to all the different evaluation intervals without any mistakes. This is in contrast to all existing end-to-end learning-based approaches such as Mena et al. (2018), which starts to make mistakes on $[0, 1]$ at 120 numbers and no longer generalises to sets drawn from $[1000, 1001]$ at 80 numbers. Vinyals et al. (2015a) already starts making mistakes on 5 numbers. Our stark improvement over existing results is evidence that the inductive biases due to the learned pairwise comparisons in our model are suitable for learning permutations, at least for this particular toy problem. In subsection C.1, we investigate what it learns that allows it to generalise this well.

Table 1: Mean squared error of image mosaic reconstruction for different datasets and number of tiles an image is split into. Lower is better. LinAssign* is the model by Mena et al. (2018), LinAssign is our reproduction of their model, PO-U and PO-LA are our models with uniform and linear assignment initialisation respectively.

| Model | MNIST | | | | CIFAR10 | | | | ImageNet $64 \times 64$ | | | |
|---|---|---|---|---|---|---|---|---|---|---|---|---|
| | $2 \times 2$ | $3 \times 3$ | $4 \times 4$ | $5 \times 5$ | $2 \times 2$ | $3 \times 3$ | $4 \times 4$ | $5 \times 5$ | $2 \times 2$ | $3 \times 3$ | $4 \times 4$ | $5 \times 5$ |
| *LinAssign** | *0.00* | *0.00* | *0.26* | *0.18* | *–* | *–* | *–* | *–* | *0.22* | *0.31* | *–* | *–* |
| LinAssign | 0.00 | 0.00 | 0.33 | 0.08 | 0.37 | 0.49 | 1.34 | 1.12 | 0.60 | 1.10 | 1.33 | 1.44 |
| PO-U | **0.00** | 0.02 | 0.46 | 0.45 | **0.11** | 0.44 | 1.23 | 1.26 | **0.14** | 0.69 | 1.20 | **1.31** |
| PO-LA | 0.00 | **0.00** | **0.07** | **0.01** | 0.18 | **0.16** | **1.07** | **0.70** | 0.16 | **0.62** | **1.13** | 1.32 |

## 4.2 RE-ASSEMBLING IMAGE MOSAICS

As a second task, we consider a problem where the model is given images that are split into $n \times n$ equal-size tiles and the goal is to re-arrange this set of tiles back into the original image. We take these images from either MNIST, CIFAR10, or a version of ImageNet with images resized down to $64 \times 64$ pixels. For this task, we use the alternative cost function described in subsection 2.4 to arrange the tiles into a grid rather than a sequence: this lets our model take relations within rows and columns into account. Again, we minimise the mean squared error to the correctly permuted image and use the Hungarian algorithm during evaluation, matching the experimental setup in Mena et al. (2018). Due to the lack of reference implementation of their model for this experiment, we use our own implementation of their model, which we verified to reproduce their MNIST results closely. Unlike them, we decide to not arbitrarily upscale MNIST images to get improved results for all models.

The mean squared errors for the different image datasets and different number of tiles an image is split into are shown in Table 1. First, notice that in essentially all cases, our model with linear assignment initialisation (PO-LA) performs best, often significantly so. On the two more complex datasets CIFAR10 and ImageNet, this is followed by our PO-U model, then the LinAssign model. We analyse what types of comparisons PO-U learns in subsection C.2.

On MNIST, LinAssign performs better than PO-U on higher tile counts because images are always centred on the object of interest. That means that many tiles only contain the background and end up completely blank; these tiles can be more easily assigned to the borders of the image by the LinAssign model than our PO-U model because the absolute position is much more important than the relative positioning to other tiles. This also points towards an issue for these cases in our cost function: because two tiles that have the same contents are treated the same by our model, it is unable to place one blank tile on one side of the image and another blank tile on the opposite side, as this would require treating the two tiles differently. This issue with backgrounds is also present on CIFAR10 to a lesser extent: notice how for the $3 \times 3$ case, the error of PO-U is much closer to LinAssign on CIFAR10 than on ImageNet, where PO-U is much better comparatively. This shows that the PO-U model is more suitable for more complex images when relative positioning matters more, and that PO-LA is able to combine the best of both methods.

## 4.3 IMPLICIT PERMUTATIONS THROUGH IMAGE CLASSIFICATION

We now turn to tasks where the goal is not producing the permutation itself, but learning a suitable permutation for a different task. For these tasks, we do not provide explicit supervision on what the permutation should be; an appropriate permutation is learned implicitly while learning to solve another task.

As the dataset, we use a straightforward modification of the image mosaic task. The image tiles are assigned to positions on a grid as before, which are then concatenated into a full image. This image is fed into a standard image classifier (ResNet-18 (He et al., 2015)) which is trained with the usual cross-entropy loss to classify the image. The idea is that the network has to learn *some* permutation of the image tiles so that the classifier can classify it accurately. This is not necessarily the permutation that restores the original image faithfully.

Table 2: Accuracy of classification from implicitly-learned image reconstructions through permutations. *max* shows the accuracy of the pre-trained model on the original images, *min* shows the accuracy of the pre-trained model on images with randomly permuted tiles.

| Model | MNIST | | | | CIFAR10 | | | | ImageNet $64 \times 64$ | | | |
|---|---|---|---|---|---|---|---|---|---|---|---|---|
| | $2 \times 2$ | $3 \times 3$ | $4 \times 4$ | $5 \times 5$ | $2 \times 2$ | $3 \times 3$ | $4 \times 4$ | $5 \times 5$ | $2 \times 2$ | $3 \times 3$ | $4 \times 4$ | $5 \times 5$ |
| *max* | *99.5* | *99.5* | *99.5* | *99.3* | *81.0* | *81.0* | *81.9* | *80.2* | *31.2* | *33.4* | *31.2* | *33.5* |
| *min* | *36.6* | *22.5* | *17.1* | *14.6* | *36.5* | *26.4* | *22.9* | *18.0* | *11.4* | *7.8* | *3.5* | *2.7* |
| LinAssign | **99.4** | 99.2 | 86.0 | 84.2 | 64.6 | 33.8 | 33.4 | **32.5** | 13.1 | 5.8 | 5.3 | 3.3 |
| PO-U | 99.3 | 98.7 | 67.9 | 69.2 | 70.8 | **41.6** | 33.3 | 29.7 | **24.6** | **12.1** | **7.3** | **5.1** |
| PO-LA | 99.3 | **99.4** | **93.3** | **89.8** | **71.6** | 40.7 | **34.2** | 32.3 | 23.4 | 10.9 | 6.3 | 4.4 |

Table 3: Accuracy on VQA v2 validation set, mean of 10 runs. The sample standard deviation over these runs is shown after the $\pm$ symbol. Overall includes the other three question categories.

| Model | Overall | Yes/No | Number | Other |
|---|---|---|---|---|
| BAN | $65.96 \pm 0.16$ | $83.34 \pm 0.09$ | $49.24 \pm 0.56$ | $57.17 \pm 0.14$ |
| BAN + LSTM | $66.06 \pm 0.13$ | $83.29 \pm 0.13$ | $49.64 \pm 0.37$ | $57.30 \pm 0.13$ |
| BAN + PO-U | $\mathbf{66.33} \pm 0.09$ | $\mathbf{83.50} \pm 0.10$ | $\mathbf{50.42} \pm 0.46$ | $\mathbf{57.48} \pm 0.10$ |

One issue with this set-up we observed is that with big tiles, it is easy for a CNN to ignore the artefacts on the tile boundaries, which means that simply permuting the tiles randomly gets to almost the same test accuracy as using the original image. To prevent the network from avoiding to solve the task, we first pre-train the CNN on the original dataset without permuting the image tiles. Once it is fully trained, we freeze the weights of this CNN and train only the permutation mechanism.

We show our results in Table 2. Generally, a similar trend to the image mosaic task with explicit supervision can be seen. Our PO-LA model usually performs best, although for ImageNet PO-U is consistently better. This is evidence that for more complex images, the benefits of linear assignment decrease (and can actually detract from the task in the case of ImageNet) and the importance of the optimisation process in our model increases. With higher number of tiles on MNIST, even though PO-U does not perform well, PO-LA is clearly superior to only using LinAssign. This is again due to the fully black tiles not being able to be sorted well by the cost function with uniform initialisation.

## 4.4 VISUAL QUESTION ANSWERING

As the last task, we consider the much more complex problem of visual question answering (VQA): answering questions about images. We use the VQA v2 dataset (Antol et al., 2015; Goyal et al., 2017), which in total contains around 1 million questions about 200,000 images from MS-COCO with 6.5 million human-provided answers available for training. We use bottom-up attention features (Anderson et al., 2018) as representation for objects in the image, which for each image gives us a *set* (size varying from 10 to 100 per image) of bounding boxes and the associated feature vector that encodes the contents of the bounding box. These object proposals have no natural ordering a priori.

We use the state-of-the-art BAN model (Kim et al., 2018) as a baseline and perform a straightforward modification to it to incorporate our module. For each element in the set of object proposals, we concatenate the bounding box coordinates, features, and the attention value that the baseline model generates. Our model learns to permute this set into a sequence, which is fed into an LSTM. We take the last cell state of the LSTM to be the representation of the set, which is fed back into the baseline model. This is done for each of the eight attention glimpses in the BAN model. We include another baseline model (BAN + LSTM) that skips the permutation learning, directly processing the set with the LSTM.

Our results on the validation set of VQA v2 are shown in Table 3. We improve on the overall performance of the state-of-the-art model by 0.37% – a significant improvement for this dataset – with 0.27% of this improvement coming from the learned permutation. This shows that there is a substantial benefit to learning an appropriate permutation through our model in order to learn better

set representations. Our model significantly improves on the number category, despite the inclusion of a counting module (Zhang et al., 2018) specifically targeted at number questions in the baseline. This is evidence that the representation learned through the permutation is non-trivial. Note that the improvement through our model is not simply due to increased model size and computation: Kim et al. (2018) found that significantly increasing BAN model size, increasing computation time similar in scale to including our model, does not yield any further gains.

## 5 DISCUSSION

In this paper, we discussed our Permutation-Optimisation module to learn permutations of sets using an optimisation-based approach. In various experiments, we verified the merit of our approach for learning permutations and, from them, set representations. We think that the optimisation-based approach to processing sets is currently underappreciated and hope that the techniques and results in this paper will inspire new algorithms for processing sets in a permutation-invariant manner. Of course, there is plenty of work to be done. For example, we have only explored one possible function for the total cost; different functions capturing different properties may be used. The main drawback of our approach is the cubic time complexity in the set size compared to the quadratic complexity of Mena et al. (2018), which limits our model to tasks where the number of elements is relatively small. While this is acceptable on the real-world dataset that we used – VQA with up to 100 object proposals per image – with only a 30% increase in computation time, our method does not scale to the much larger set sizes encountered in domains such as point cloud classification. Improvements in the optimisation algorithm may improve this situation, perhaps through a divide-and-conquer approach.

We believe that going beyond tensors as basic data structures is important for enabling higher-level reasoning. As a fundamental mathematical object, sets are a natural step forward from tensors for modelling unordered collections. The property of permutation invariance lends itself to greater abstraction by allowing data that has no obvious ordering to be processed, and we took a step towards this by learning an ordering that existing neural networks are able to take advantage of.

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

## A  PSEUDOCODE OF ALGORITHM

---

**Algorithm 1** Forward pass of permutation-optimisation algorithm

---

1: **Input**: $\boldsymbol{X} \in \mathbb{R}^{N \times M}$ with $\boldsymbol{x}_i$ as rows in arbitrary order
2: **Learnable parameters**: weights that parametrise $F$, step size $\eta$
3:
4: $C_{ij} \leftarrow \text{normed}(F(\boldsymbol{x}_i, \boldsymbol{x}_j))$         ▷ compute ordering costs (equation 10)
5: initialise $\widetilde{\boldsymbol{P}}$          ▷ either uniform or linear assignment init (equation 6)
6: **for** t $\leftarrow 1, T$ **do**
7:    $\boldsymbol{P} \leftarrow S(\widetilde{\boldsymbol{P}})$       ▷ normalise assignment with Sinkhorn operator (Appendix B)
8:    $\boldsymbol{G} \leftarrow \partial c(\boldsymbol{P})/\partial \boldsymbol{P}$     ▷ compute gradient of normalised assignment (equation 4)
9:    $\widetilde{\boldsymbol{P}} \leftarrow \widetilde{\boldsymbol{P}} - \eta \boldsymbol{G}$    ▷ gradient descent step on unnormalised assignment (equation 7)
10: **end for**
11: $\boldsymbol{P} \leftarrow S(\widetilde{\boldsymbol{P}})$
12: $\boldsymbol{Y} \leftarrow \boldsymbol{PX}$           ▷ permute rows of $\boldsymbol{X}$ to obtain output $\boldsymbol{Y}$

---

## B  SINKHORN OPERATOR

The Sinkhorn operator $S$ as defined in Adams & Zemel (2011) is:

$$\mathcal{T}_r(\boldsymbol{X})_{ij} = X_{ij} / \sum_k X_{ik} \tag{11}$$

$$\mathcal{T}_c(\boldsymbol{X})_{ij} = X_{ij} / \sum_k X_{kj} \tag{12}$$

$$S^{(0)}(\boldsymbol{X}) = \exp(\boldsymbol{X}) \tag{13}$$

$$S^{(l+1)}(\boldsymbol{X}) = \mathcal{T}_c(\mathcal{T}_r(S^{(l)}(\boldsymbol{X}))) \tag{14}$$

$$S(\boldsymbol{X}) = S^{(L)}(\boldsymbol{X}) \tag{15}$$

$\mathcal{T}_r$ normalises each row, $\mathcal{T}_c$ normalises each column of a square matrix $\boldsymbol{X}$ to sum to one. This formulation is different from the normal Sinkhorn operator by Sinkhorn (1964) by exponentiating all entries first and running for a fixed number of steps $L$ instead of for steps approaching infinity. Mena et al. (2018) include a temperature parameter on the exponentiation, which acts analogously to temperature in the softmax function. In this paper, we fix $L$ to 4.

## C  QUALITATIVE ANALYSIS OF LEARNED COMPARISON FUNCTIONS

### C.1  NUMBER SORTING

First, we investigate what comparison function $F$ is learned for the number sorting task. We start with plotting the outputs of $F$ for different pairs of inputs in Figure 2. From this, we can see that it learns a sensible comparison function where it outputs a negative number when the first argument is lower than the second, and a positive number vice versa.

The easiest way to achieve this is to learn $f(x_i, x_j) = x_i$, which results in $F(x_i, x_j) = x_i - x_j$. By plotting the outputs of the learned $f$ in Figure 3 we can see that something close to this has indeed been learned. The learned $f$ mostly depends on the second argument and is a scaled and shifted version of it. It has not learned to completely ignore the first argument, but the deviations from it are small enough that the cost function of the permutation is able to compensate for it. We can see that there is a faint grey diagonal area going from $(0, 0)$ to $(1, 1)$ and to $(1000, 1000)$, which could be an artifact from $F$ having small gradients due to its skew-symmetry when two numbers are close to each other.

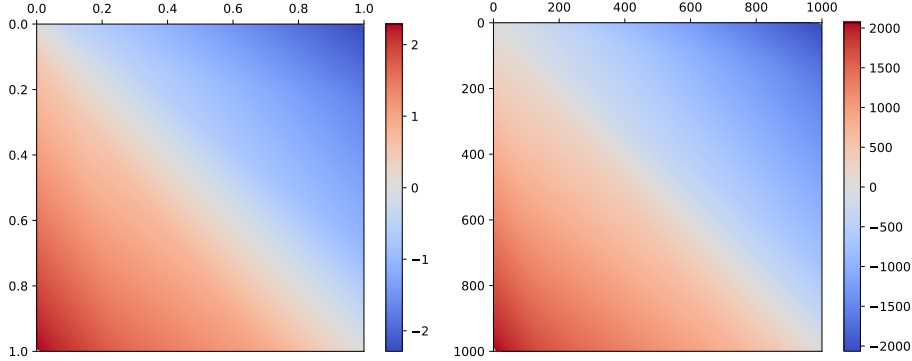

Figure 2: Outputs of $F$ for different pairs of numbers as input. Red indicates that the number on the left should be ordered after the number at the top, blue indicates the opposite. Evaluation intervals are $[0, 1]$ (left) and $[0, 1000]$ (right).

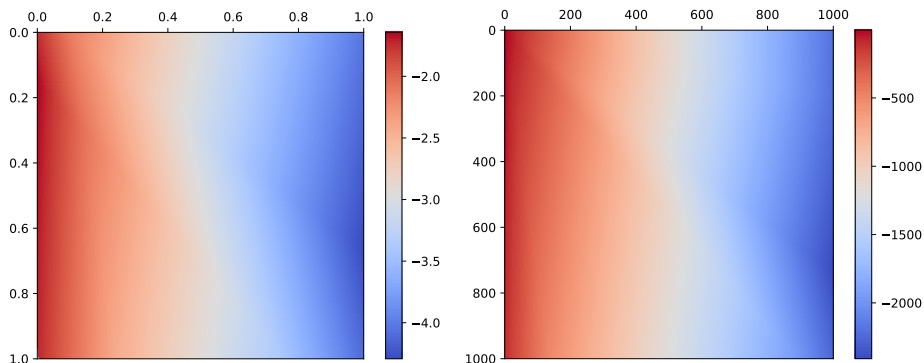

Figure 3: Outputs of $f$ for different pairs of numbers as input. Evaluation intervals are $[0, 1]$ (left) and $[0, 1000]$ (right).

## C.2   IMAGE MOSAICS

Next, we investigate the behaviour of $F$ on the image mosaic task. Since our model uses the outputs of $F$ in the optimisation process, we find it easier to interpret $F$ over $f$ in the subsequent analysis.

### C.2.1   OUTPUTS

We start by looking at the output of $F_1$ (costs for left-to-right ordering) and $F_2$ (costs for top-to-bottom ordering) for MNIST $2 \times 2$, shown in Figure 4. First, there is a clear entry in each row and column of both $F_1$ and $F_2$ that has the highest absolute cost (high colour saturation) whenever the corresponding tiles fit together correctly. This shows that it successfully learned to be confident what order two tiles should be in when they fit together. From the two 2-by-2 blocks of red and blue on the anti-diagonal, we can also see that it has learned that for the per-row comparisons ($F_1$), the tiles that should go into the left column should generally compare to less than (i.e. should be permuted to be to the left of) the tiles that go to the right. Similarly, for the per-column comparisons ($F_2$) tiles that should be at the top compare to less than tiles that should be at the bottom. Lastly, $F_1$ has a low absolute cost when comparing two tiles that belong in the same column. These are the entries in the matrix at the coordinates $(1, 2)$, $(2, 1)$, $(4, 3)$, and $(3, 4)$. This makes sense, as $F_1$ is concerned with whether one tile should be to the left or right of another, so tiles that belong in the same column should not have a preference either way. A similar thing applies to $F_2$ for tiles that belong in the same column.

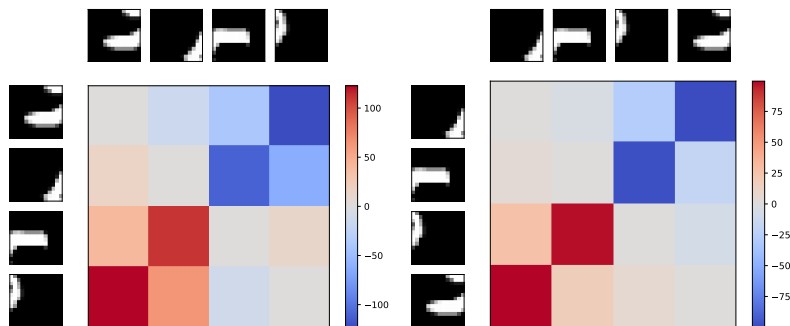

Figure 4: Outputs of $F_1$ (left half, row comparisons) and $F_2$ (right half, column comparisons) for pairs of tiles from an image in MNIST. For $F_1$, the tiles are sorted left-to-right if only $F_1$ was used as cost. For $F_2$, the tiles are sorted top-to-bottom if only $F_2$ was used as cost. Blue indicates that the tile to the left of this entry should be ordered left of the tile at the top for $F_1$, the tile on the left should be ordered above the tile at the top for $F_2$. The opposite applies for red. The saturation of the colour indicates how strong this ordering is.



Figure 5: Sensitivity to positions within a tile for MNIST $2 \times 2$ (left), $3 \times 3$ (middle), and $4 \times 4$ (right). The left plot of each pair shows $F_1$, the right plot shows $F_2$.

### C.2.2 SENSITIVITY TO POSITIONS

Next, we investigate what positions within the tiles $F_1$ and $F_2$ are most sensitive to. This illustrates what areas of the tiles are usually important for making comparisons. We do this by computing the gradients of the absolute values of $F$ with respect to the input tiles and averaging over many inputs. For MNIST $2 \times 2$ (Figure 5, left), it learns no particular spatial pattern for $F_1$ and puts slightly more focus away from the centre of the tile for $F_2$. As we will see later, it learns something that is very content-dependent rather than spatially-dependent. With increasing numbers of tiles on MNIST, it tends to focus more on edges, and especially on corners. For the CIFAR10 dataset (Figure 6), there is a much clearer distinction between left-right comparisons for $F_1$ and top-bottom comparisons for $F_2$. For the $2 \times 2$ and $4 \times 4$ settings, it relies heavily on the pixels on the left and right borders for left-to-right comparisons, and top and bottom edges for top-to-bottom comparisons. Interestingly, $F_1$ in the $3 \times 3$ setting (middle pair) on CIFAR10 focuses on the left and right halves of the tiles, but specifically avoids the borders. A similar thing applies to $F_2$, where a greater significance is given to pixels closer to the middle of the image rather than only focusing on the edges. This suggests that it learns to not only match up edges as with the other tile numbers, but also uses the content within the tile to do more sophisticated content-based comparisons.

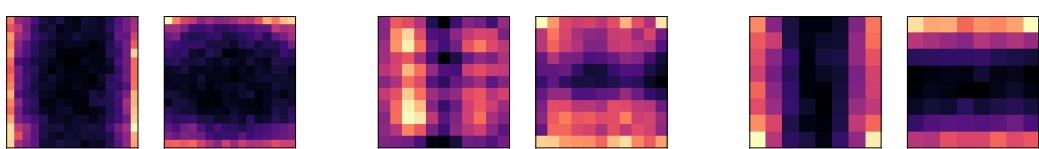

Figure 6: Sensitivity to positions within a tile of the comparisons for CIFAR10 $2 \times 2$ (left), $3 \times 3$ (middle), and $4 \times 4$ (right). The left plot of each pair shows $F_1$, the right plot shows $F_2$.

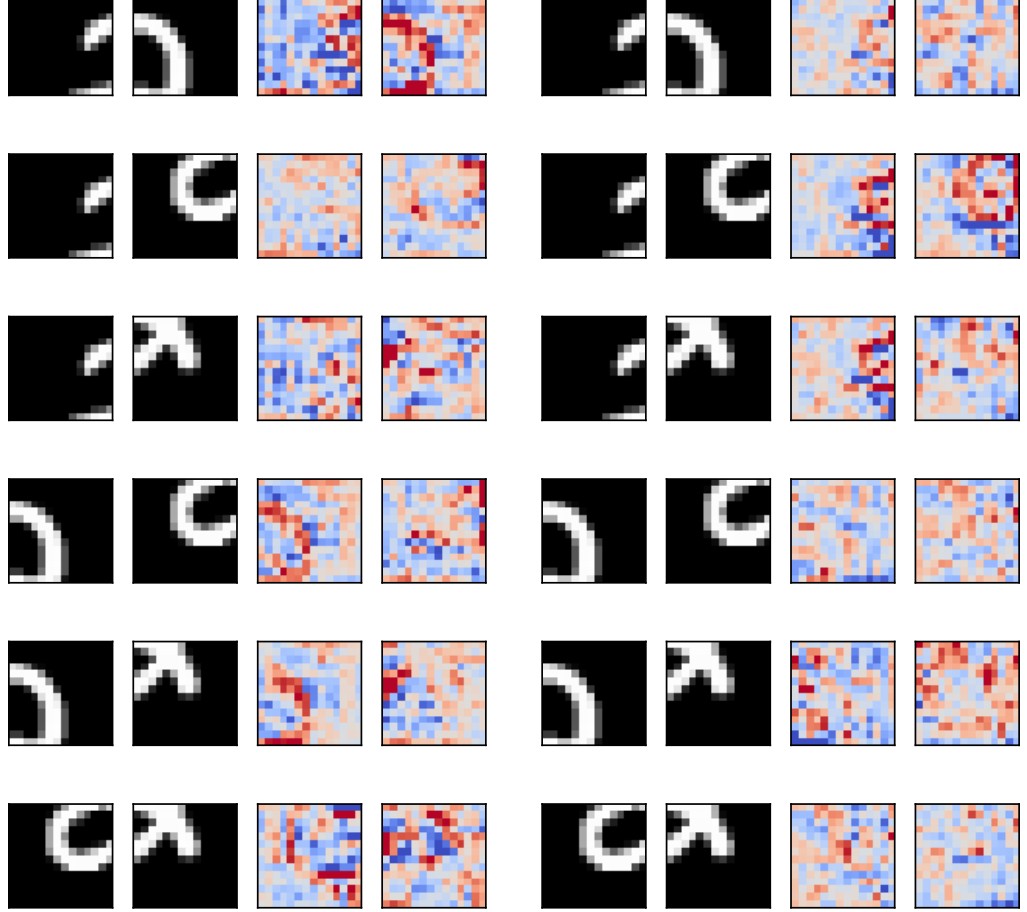

Figure 7: Gradient maps of pairs of tiles from MNIST for $F_1$ (left half) and $F_2$ (right half). Each group of four consists of: tile 1, tile 2, gradient of $F(t_1, t_2)$ with respect to tile 1, gradient of $F(t_2, t_1)$ with respect to tile 2.

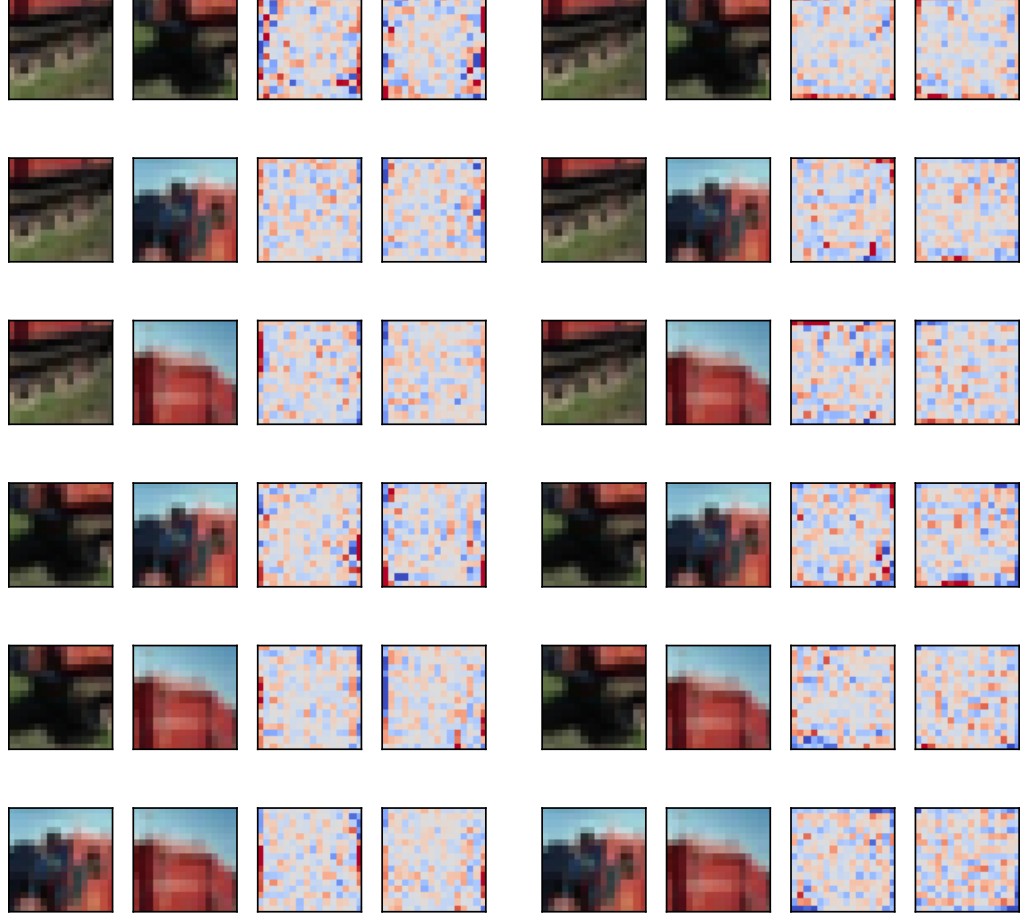

Figure 8: Gradient maps of pairs of tiles from CIFAR10 for $F_1$ (left half) and $F_2$ (right half). Each group of four consists of: tile 1, tile 2, gradient of $F(t_1, t_2)$ with respect to tile 1, gradient of $F(t_2, t_1)$ with respect to tile 2.

### C.2.3 Per-tile gradients

Lastly, we can look at the gradients of $F$ with respect to the input tiles for specific pairs of tiles, shown in Figure 7 and Figure 8. This gives us a better insight into what changes to the input tiles would affect the cost of the comparison the most. These figures can be understood as follows: for each pair of tiles, we have the corresponding two gradient maps next to them. Brightening the pixels for the blue entries in these gradient maps would order the corresponding tile more strongly towards the left for $F_1$ and towards the top for $F_2$. The opposite applies to brightening the pixels with red entries. Vice versa, darkening pixels with blue entries orders the tile more strongly towards the right for $F_1$ and the bottom for $F_2$. More saturated colours in the gradient maps correspond to greater effects on the cost when changing those pixels.

We start with gradients on the tiles for an input showing the digit 2 on MNIST $2 \times 2$ in Figure 7. We focus on the first row, left side, which shows a particular pair of tiles from this image and their gradients of $F_1$ (left-to-right ordering), and we share some of our observations here:

- The gradients of the second tile show that to encourage the permutation to place it to the right of the first tile, it is best to increase the brightness of the curve in tile 2 that is already white (red entries in tile 2 gradient map) and decrease the black pixels around it (blue entries). This means that it recognised that this type of curve is important in determining that it should be placed to the right, perhaps because it matches up with the start of the curve from tile 1. We can imagine the curve in the gradient map of tile 2 roughly forming part of a 7 rather than a 2 as well, so it is not necessarily looking for the curve of a 2 specifically.

- In the gradient map of the first tile, we can see that to encourage it to be placed to the left of tile 2, increasing the blue entries would form a curve that would make the first tile look like part of an 8 rather than a 2, completing the other half of the curve from tile 2. This means that it has learned that to match something with the shape in tile 2, a loop that completes it is best, but the partial loop that we have in tile 1 satisfies part of this too.

- Notice how the gradient of tile 1 changes quite a bit when going from row 1 to row 3, where it is paired up with different tiles. This suggests that the comparison has learned something about the specific comparison between tiles being made, rather than learning a general trend of where the tile should go. The latter is what a linear assignment model is limited to doing because it does not model pairwise interactions.

- In the third row, we can see that even though the two tiles do not match up, there is a red blob on the left side of the tile 2 gradient map. This blob would connect to the top part of the line in tile 1, so it makes sense that making the two tiles match up more on the border would encourage tile 2 to be ordered to the right of tile 1.

Similar observations apply to the right half of Figure 7, such as row 5, where tile 1 (which should go above tile 2) should have its pixels in the bottom left increased and tile 2 should have its pixels in the top left increased in order for tile 1 to be ordered before (i.e. above) tile 2 more strongly.

On CIFAR10 $2 \times 2$ in Figure 8, it is enough to focus on the borders of the tiles. Here, it is striking how specifically it tries to match edge colours between tiles. For example, consider the blue sky in the left half ($F_1$), row 6. To order tile 1 to the left of tile 2, we should change tile 1 to have brighter sky and darker red on the right border, and also darken the black on the left border so that it matches up less well with the right border of tile 2, where more of the bright sky is visible. For tile 2, the gradient shows that it should also match up more on the left border, and have increase the amount of bright pixels, i.e. sky, on the right border, again so that it matches up less well with the left border of tile 1 if they were to be ordered the opposing way.

# D  JUSTIFICATION FOR ALTERNATIVE UPDATE

First, the gradient of $S(\boldsymbol{X})$ is:

$$\frac{\partial S(\boldsymbol{X})}{\partial \boldsymbol{X}} = \frac{\partial \exp(\boldsymbol{X})}{\partial \boldsymbol{X}} \frac{\partial \mathcal{T}_r(\exp(\boldsymbol{X}))}{\partial \exp(\boldsymbol{X})} \frac{\partial \mathcal{T}_c(\mathcal{T}_r(\exp(\boldsymbol{X})))}{\partial \mathcal{T}_r(\exp(\boldsymbol{X}))} \cdots \tag{16}$$

$$\frac{\partial \mathcal{T}_r(\boldsymbol{X})_{uv}}{\partial X_{ij}} = \mathbb{1}_{u=i} \frac{\mathbb{1}_{v=j} \sum_k X_{uk} - X_{uv}}{(\sum_k X_{uk})^2} \tag{17}$$

$$\frac{\partial \mathcal{T}_c(\boldsymbol{X})_{uv}}{\partial X_{ij}} = \mathbb{1}_{v=j} \frac{\mathbb{1}_{u=i} \sum_k X_{kv} - X_{uv}}{(\sum_k X_{kv})^2} \tag{18}$$

where $\mathbb{1}$ is the indicator function that returns 1 if the condition is true and 0 otherwise.

We compared the entropy of the permutation matrices obtained with and without using the "proper" gradient with $\partial S(\widetilde{\boldsymbol{P}})/\partial \widetilde{\boldsymbol{P}}$ as term in it and found that our version has a significantly lower entropy. To understand this, it is enough to focus on the first two terms in equation 16, which is essentially the gradient of a softmax function applied row-wise to $\boldsymbol{P}$.

Let $\boldsymbol{x}$ be a row in $\boldsymbol{P}$ and $s_i$ be the $i$th entry in the softmax function applied to $\boldsymbol{x}$. Then, the gradient is:

$$\frac{\partial s_i}{\partial x_j} = s_i(\mathbb{1}_{i=j} - s_j) \tag{19}$$

Since this is a product of entries in a probability distribution, the gradient vanishes quickly as we move towards a proper permutation matrix (all entries very close to 0 or 1). By using our alternative update and thus removing this term from our gradient, we can avoid the vanishing gradient problem.

Gradient descent is not efficient when the gradient vanishes towards the optimum and the optimum – in our case a permutation matrix with exact ones and zeros as entries – is infinitely far away. Since we prefer to use a small number of steps in our algorithm for efficiency, we want to reach a good solution as quickly as possible. This justifies effectively ignoring the step size that the gradient suggests and simply taking a step in a similar direction as the gradient in order to be able to saturate the Sinkhorn normalisation sufficiently, thus obtaining a doubly stochastic matrix that is closer to a proper permutation matrix in the end.

# E  QUADRATIC PROGRAMMING PROBLEM

We can write our total cost function as a quadratic program in the standard $\boldsymbol{x}^T \boldsymbol{Q} \boldsymbol{x}$ form with linear constraints. We leave out the constraints here as they are not particularly interesting. First, we can define $\boldsymbol{O} \in \mathbb{R}^{N \times N}$ as:

$$O_{kk'} = \begin{cases} -1 & \text{if } k > k' \\ 0 & \text{if } k = k' \\ 1 & \text{if } k < k' \end{cases} \tag{20}$$

and with it, $\boldsymbol{Q} \in \mathbb{R}^{N^2 \times N^2}$ as:

$$Q_{(ik)(jk')} = C_{ij} O_{kk'} \tag{21}$$

Then we can write the cost function as:

$$c(\boldsymbol{P}) = \sum_{ij} C_{ij} \sum_{kk'} P_{ik} P_{jk'} O_{kk'} \tag{22}$$

$$= \sum_{ik} \sum_{jk'} P_{ik} (C_{ij} O_{kk'}) P_{jk'} \tag{23}$$

$$= \sum_{(ik)} \sum_{(jk')} P_{(ik)} Q_{(ik)(jk')} P_{(jk')} \tag{24}$$

$$= \boldsymbol{p}^T \boldsymbol{Q} \boldsymbol{p} \tag{25}$$

where there is some bijection between a pair of indices $(i, k)$ and the index $l$ and $\boldsymbol{p}$ is a flattened version of $\boldsymbol{P}$ with $p_l = P_{ik}$. $\boldsymbol{Q}$ is indefinite because the total cost can be negative: a uniform initialisation for $\boldsymbol{P}$ has a cost of 0, better permutations have negative cost, worse permutations have positive cost. Thus, the problem is non-convex and the problem is possibly NP-hard. Also, since we have flattened $\boldsymbol{P}$ into $\boldsymbol{p}$, the number of optimisation variables is quadratic in the set size $N$. Even if this were a convex quadratic program, methods such as OptNet (Amos & Kolter, 2017) have cubic time complexity in the number of optimisation variables, which makes it $O(N^6)$ for our case.

## F EXPERIMENTAL DETAILS

All of our experiments can be reproduced using our implementation at `https://github.com/Cyanogenoid/perm-optim` in PyTorch (Paszke et al., 2017) through the `experiments/all.sh` script. For the former three experiments, we use the following hyperparameters throughout:

- Optimiser: Adam (Kingma & Ba, 2015) (default settings in PyTorch: $\beta_1 = 0.9, \beta_2 = 0.999, \epsilon = 10^{-8}$)
- Initial step size $\eta$ in inner gradient descent: 1.0

All weights are initialised with Xavier initialisation (Glorot & Bengio, 2010). We choose the $f$ within the ordering cost function $F$ to be a small MLP. The input to $f$ has 2 times the number of dimensions of each element, obtained by concatenating the pair of elements. This is done for all pairs that can be formed from the input set. This is linearly projected to some number of hidden units to which a ReLU activation is applied. Lastly, this is projected down to 1 dimension for sorting numbers and VQA, and 2 dimensions for assembling image mosaics (1 output for row-wise costs, 1 output for column-wise costs). These outputs are used for creating the ordering cost matrix $\boldsymbol{C}$.

### F.1 SORTING NUMBERS

- Inner gradient descent steps $T$: 6
- Adam learning rate: 0.1
- Batch size: 512
- Number of sets to sort in training set: $2^{18}$
- Set sizes: 5, 10, 15, 80, 100, 120, 512, 1024
- Evaluation intervals: $[0, 1], [0, 10], [0, 1000], [1, 2], [10, 11], [100, 101], [1000, 1001]$ (same as in Mena et al. (2018))
- $F$ size of hidden dimension: 16

The ordering cost function $F$ concatenates the two floats of each pair and applies a 2-layer MLP that takes the 2 inputs to 16 hidden units, ReLU activation, then to one output.

For evaluation, we switch to double precision floating point numbers. This is because for the interval $[1000, 1001]$, as the set size increases, there are not enough unique single precision floats in that interval for the sets to contain only unique floats with high probability (the birthday problem). Using double precision floats avoids this issue. Note that using single precision floats is enough for the other intervals and smaller set sizes, and training is always done on the interval $[0, 1]$ at single precision.

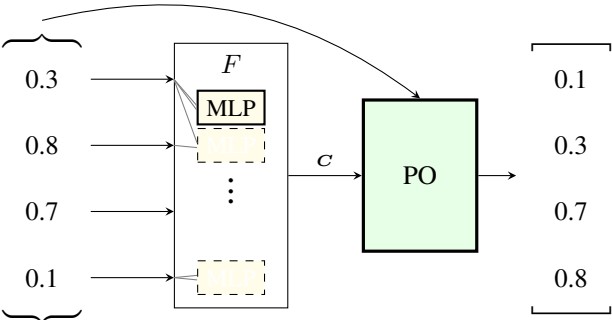

Figure 9: Network architecture for number sorting. The MLP in $F$ computes $f$ and is shared across pairs of set elements. The PO block performs the optimisation with the given costs and permutes the input set.

## F.2 Re-assembling Image Mosaics

- Adam learning rate: $10^{-3}$
- Inner gradient descent steps $T$: 4
- Batch size: 32
- Training epochs: 20 (MNIST, CIFAR10) or 1 (ImageNet)
- $F$ size of hidden dimension: 64 (MNIST, CIFAR10) or 128 (ImageNet)

For all three image datasets from which we take images (MNIST, CIFAR10, ImageNet), we first normalise the inputs to have zero mean and standard deviation one over the dataset as is common practice. For ImageNet, we crop rectangular images to be square by reducing the size of the longer side to the length of the shorter side (centre cropping). Images that are not exactly divisible by the number of tiles are first rescaled to the nearest bigger image size that is exactly divisible. Following Mena et al. (2018), we process each tile with a $5 \times 5$ convolution with padding and stride 1, $2 \times 2$ max pooling, and ReLU activation. This is flattened into a vector to obtain the feature vector for each tile, which is then fed into our $F$. Unlike Mena et al. (2018), we decide not to arbitrarily upscale MNIST images by a factor of two, even when upscaling results in slightly better performance in general.

While we were able to mostly reproduce their MNIST results, we were not able to reproduce their ImageNet results for the $3 \times 3$ case. In general, we observed that good settings for their model also improved the results of our PO-U and PO-LA models. Better hyperparameters than what we used should improve all models similarly while keeping the ordering of how well they perform the same.

This task is also known as jigsaw puzzle (Noroozi & Favaro, 2016), but we decided on naming it image mosaics because the tiles are square which can lead to multiple solutions, rather than the typical unique solution in traditional jigsaw puzzles enforced by the different tile shapes.

## F.3 Implicit Permutations through Image Classification

We use the same setting as for the image mosaics, but further process the output image with a ResNet-18. For MNIST and CIFAR10, we replace the first convolutional layer with one that has a $3 \times 3$ kernel size and no striding. This ResNet-18 is first trained on the original dataset for 20 epochs (1 for ImageNet), though images may be rescaled if the image size is not divisible by the number of tiles per side. All weights are then frozen and the permutation method is trained for 20 epochs (1 for ImageNet). As stated previously, this is necessary in order for the ResNet-18 to not use each tile individually and ignore the resulting artefacts from the permuted tiles. This is also one of the reasons why we downscale ImageNet images to $64 \times 64$ pixels. Because the resulting image tiles are so big while the receptive field of ResNet-18 is relatively small if we were to use $256 \times 256$ images, the permutation artefacts barely affect results because they are only a small fraction of the globally-pooled features. The permutation permutes each set of tiles, which are reconstructed (without use of the Hungarian algorithm) into an image, which is then processed by the ResNet-18.

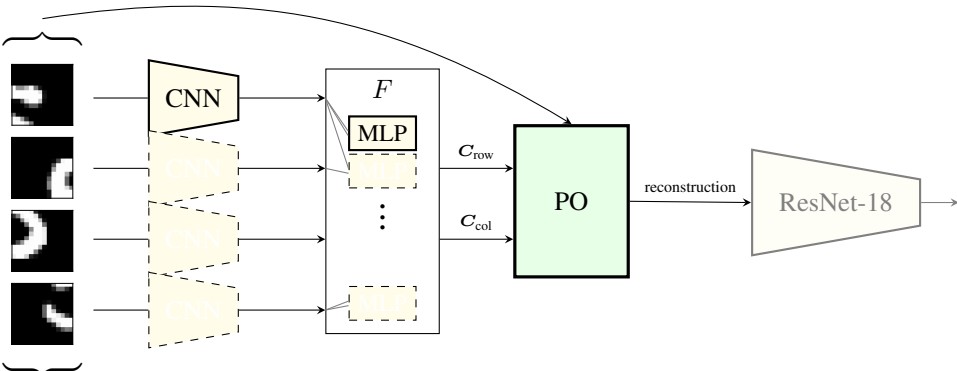

Figure 10: Network architecture for image mosaic tasks. The small CNN and the MLP in $F$ is shared across set elements and pairs of set elements respectively. The PO block performs the optimisation with the given row and column costs and permutes the input set. The ResNet-18 network at the end is only present in the implicit permutation setting.

We observed that the LinAssign model by Mena et al. (2018) consistently results in NaN values after Sinkhorn normalisation in this set-up, despite our Sinkhorn implementation using the numerically-stable version of softmax with the exp-normalise trick. We avoided this issue by clipping the outputs of their model into the [-10, 10] interval before Sinkhorn normalisation. We did not observe these NaN issues with our PO-U model.

### F.4 VISUAL QUESTION ANSWERING

We use the official implementation of BAN as baseline without changing any of the hyperparameters. We thus refer to Kim et al. (2018) for details of their model architecture and hyperparameters. The only change to hyperparameters that we make is reducing the batch size from 256 to 112 due to the GPU memory requirements of the baseline model, even without our permutation mechanism.

The BAN model generates attention weights between all object proposals in a set and words of the question. We take the attention weight for a single object proposal to be the maximum attention weight for that proposal over all words of the question, the same as in their integration of the counting module. Each element of the set, corresponding to object proposals, is the concatenation of this attention logit, bounding box coordinates, and the feature vector projected from 2048 down to 8 dimensions. We found this projection necessary to not inhibit learning of the rest of the model, which might be due to gradient clipping or other hyperparameters that are no longer optimal in the BAN model. This set of object proposals is then permuted with $T = 3$ and a 2-layer MLP with hidden dimension 128 for $f$ to produce the ordering costs. The elements in the permuted sequence are weighted by how relevant each proposal is (sigmoid of the corresponding attention logit) and the sequence is then fed into an LSTM with 128 units. The last cell state of the LSTM is the set representation which is projected, ReLUd, and added back into the hidden state of the BAN model. The remainder of the BAN model is now able to use information from this set representation. There are 8 attention glimpses, so we process each of these with a PO-U module and an LSTM with shared parameters across these 8 glimpses.

### G ADDITIONAL RESULTS

An interesting aspect we observed throughout all experiments is how the learned step size $\eta$ changes during training. At the start of training, it decreases from its initial value of 1, thus reducing the influence of the permutation mechanism. Then, $\eta$ starts rising again, usually ending up at a value above 1 at the end of training. This can be explained by the ordering cost being very inaccurate at the start of training, since it has not been trained yet. Through training, the ordering cost improves and it becomes more beneficial for the influence of the PO module on the permutation to increase.

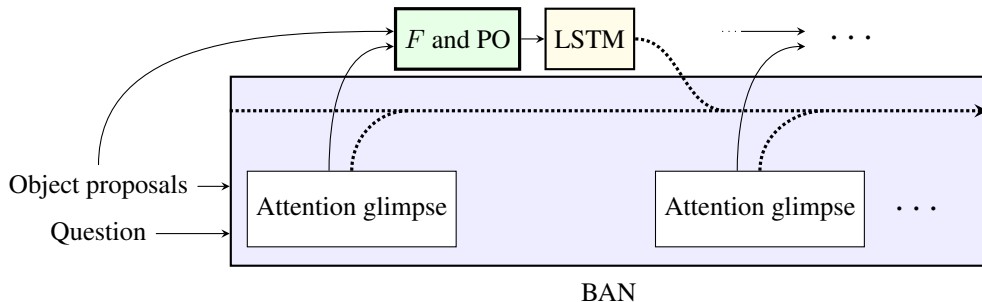

Figure 11: Network architecture for visual question answering task using BAN with 8 glimpses (2 shown) as baseline. We add a shared PO-U module with an LSTM to process its output for each glimpse. The outputs of the BAN attention and the LSTM are added into the hidden state of the BAN network.

## G.1 RE-ASSEMBLING IMAGE MOSAICS

In Table 4, we show the accuracy corresponding to the results in Table 1 where the permutation has been trained with explicit supervision. In Figure 12, Figure 13, and Figure 14, we show some example reconstructions that have been learnt by our PO-U model. Starting from a uniform assignment at the top, the figures show reconstructions as a permutation is being optimised. Generally, it is able to reconstruct most images fairly well. Due to the poor quality of many of these reconstructions (particularly on ImageNet), the last two figures show reconstructions on the $2 \times 2$ versions of the datasets rather than $3 \times 3$.

Table 4: Accuracy of image mosaic reconstruction. Higher is better. A permutation is considered correct if all tiles are placed correctly. Because of indistinguishable tiles at higher tile counts (for example multiple completely blank tiles on MNIST) it becomes very unlikely to guess the correct ground-truth matching at higher tile counts. LinAssign* results come from Mena et al. (2018).

| Model | MNIST | | | | CIFAR10 | | | | ImageNet $64 \times 64$ | | | |
|---|---|---|---|---|---|---|---|---|---|---|---|---|
| | $2 \times 2$ | $3 \times 3$ | $4 \times 4$ | $5 \times 5$ | $2 \times 2$ | $3 \times 3$ | $4 \times 4$ | $5 \times 5$ | $2 \times 2$ | $3 \times 3$ | $4 \times 4$ | $5 \times 5$ |
| *LinAssign** | *100* | *72* | *3* | *0* | *–* | *–* | *–* | *–* | *81* | *47* | *0* | *0* |
| LinAssign | 99.7 | 66.9 | 0.8 | 0.0 | 68.8 | 30.3 | 0.0 | 0.0 | 47.7 | 4.0 | 0.0 | **0.0** |
| PO-U | **100.0** | 65.9 | 0.2 | 0.0 | 86.2 | 34.4 | 0.0 | 0.0 | **85.9** | 19.2 | 0.1 | 0.0 |
| PO-LA | 99.9 | **73.1** | **1.8** | **0.0** | **87.3** | **66.0** | **0.6** | **0.3** | 84.2 | **28.6** | **0.1** | 0.0 |

Table 5: Mean squared error of implicitly-learned reconstruction. Lower is better.

| Model | MNIST | | | | CIFAR10 | | | | ImageNet $64 \times 64$ | | | |
|---|---|---|---|---|---|---|---|---|---|---|---|---|
| | $2 \times 2$ | $3 \times 3$ | $4 \times 4$ | $5 \times 5$ | $2 \times 2$ | $3 \times 3$ | $4 \times 4$ | $5 \times 5$ | $2 \times 2$ | $3 \times 3$ | $4 \times 4$ | $5 \times 5$ |
| *max* | *0.00* | *0.00* | *0.00* | *0.00* | *0.00* | *0.00* | *0.00* | *0.00* | *0.00* | *0.00* | *0.00* | *0.00* |
| *min* | *1.59* | *1.63* | *1.91* | *1.72* | *1.76* | *1.91* | *2.11* | *2.01* | *1.48* | *1.72* | *1.79* | *1.83* |
| LinAssign | 0.02 | 0.05 | 0.73 | 1.11 | 0.56 | 1.29 | 1.53 | 1.62 | 0.88 | 1.33 | 1.32 | **1.39** |
| PO-U | 0.02 | 0.15 | 1.38 | 1.24 | 0.33 | 1.03 | **1.44** | **1.34** | 0.44 | 1.16 | **1.28** | 1.47 |
| PO-LA | **0.01** | **0.03** | **0.50** | **0.93** | **0.28** | **1.00** | 1.47 | 1.55 | **0.41** | **1.15** | 1.41 | 1.43 |

## G.2 IMPLICIT PERMUTATIONS THROUGH IMAGE CLASSIFICATION

In Table 5, we show the mean squared error reconstruction loss corresponding to the results in Table 2. These show similar trends as before. In Figure 15, Figure 16, and Figure 17, we show some example reconstructions that have been learnt by our PO-U model on $3 \times 3$ versions of the image datasets.

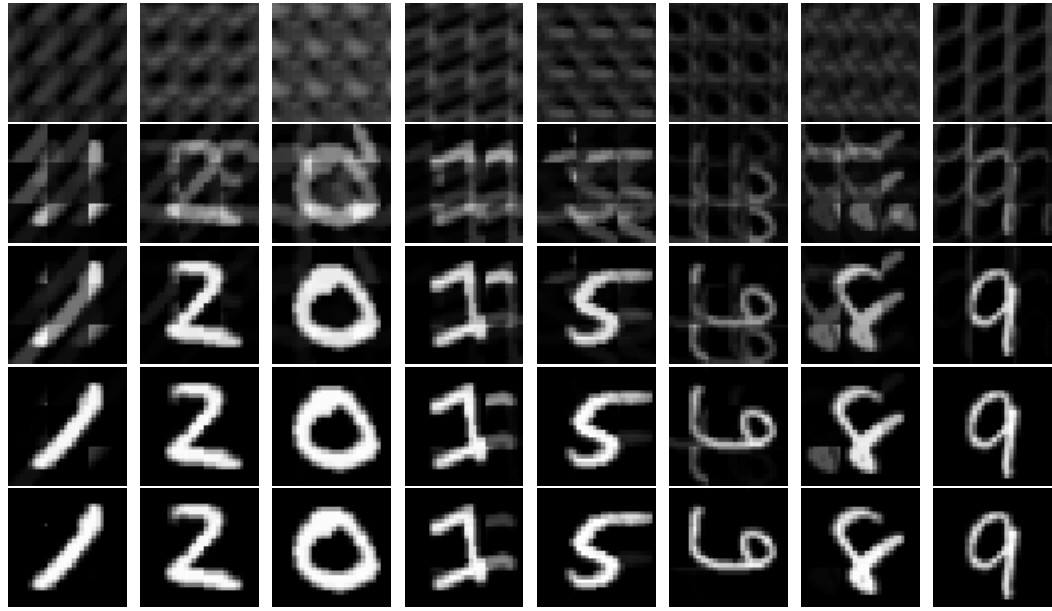

Figure 12: Example reconstructions of PO-U as they are being optimised on MNIST $3 \times 3$ with explicit supervision. These examples have not been cherry-picked.

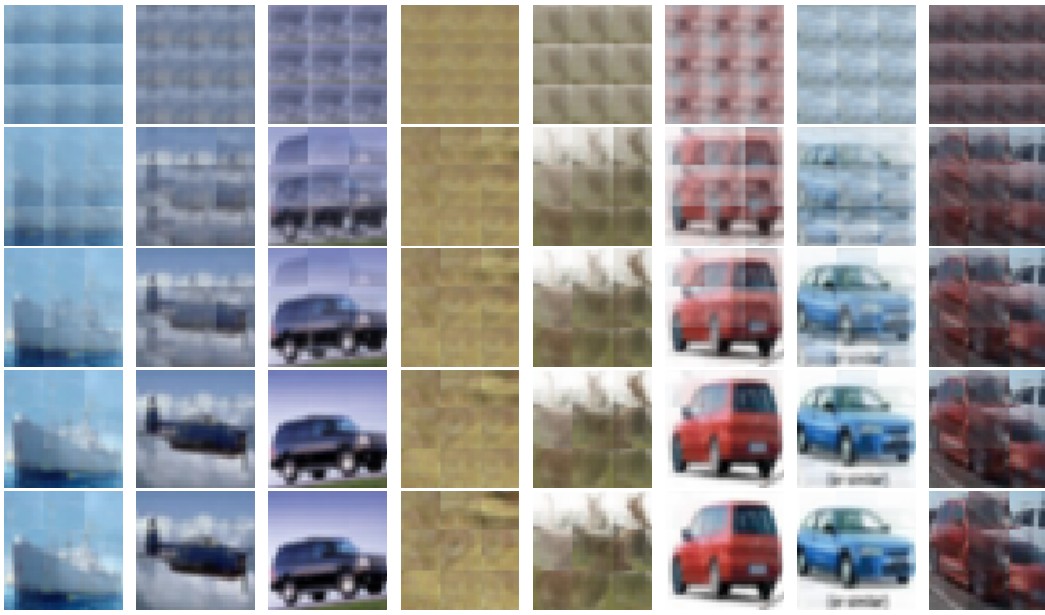

Figure 13: Example reconstructions of PO-U as they are being optimised on CIFAR10 $3 \times 3$ with explicit supervision. These examples have not been cherry-picked.

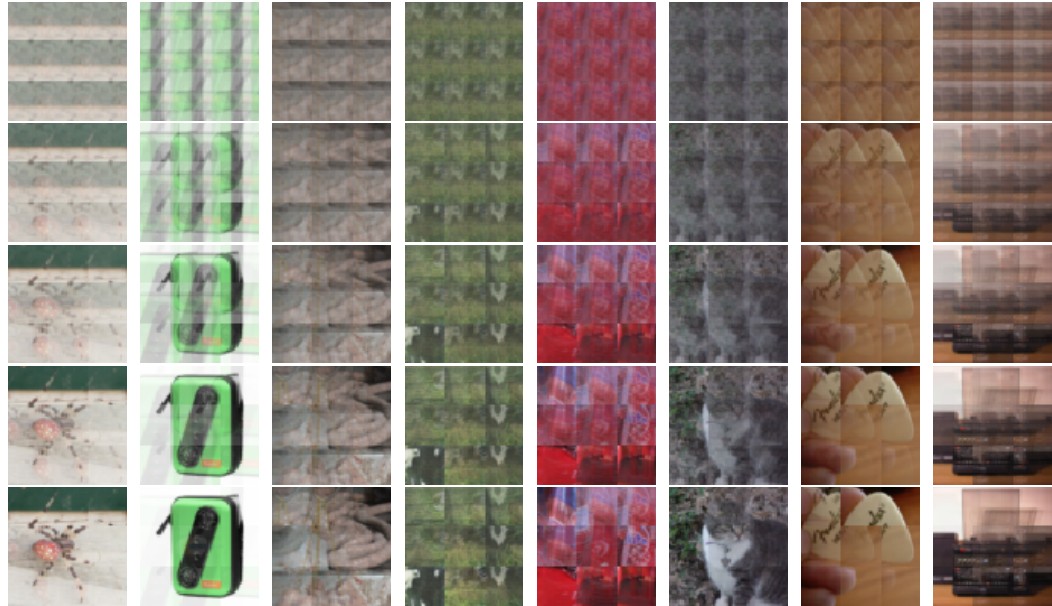

Figure 14: Example reconstructions of PO-U as they are being optimised on ImageNet $3 \times 3$ with explicit supervision. These examples have not been cherry-picked.

Because the quality of implicit CIFAR10 and ImageNet reconstructions are relatively poor, we also include Figure 18, and Figure 19 on $2 \times 2$ versions. Starting from a uniform assignment at the top, the figures show reconstructions as a permutation is being optimised. The reconstructions here are clearly noisier than before due the supervision only being implicit. This is evidence that while our method is superior to existing methods in terms of reconstruction error and accuracy of the classification, there is still plenty of room for improvement to allow for better implicitly learned permutations. Keep in mind that it is not necessary for the permutation to produce the original image exactly, as long as the CNN can consistently recognise what the permutation method has learned. Our models tend to naturally learn reconstructions that are more similar to the original image than the LinAssign model.

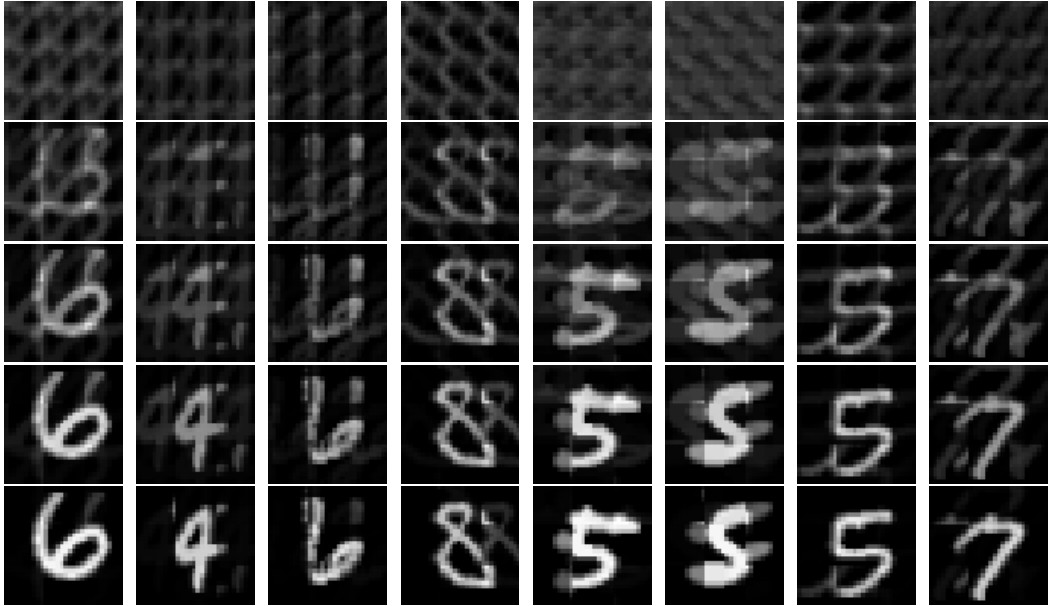

Figure 15: Example reconstructions of PO-U as they are being optimised on MNIST $3 \times 3$ with implicit supervision. These examples have not been cherry-picked.

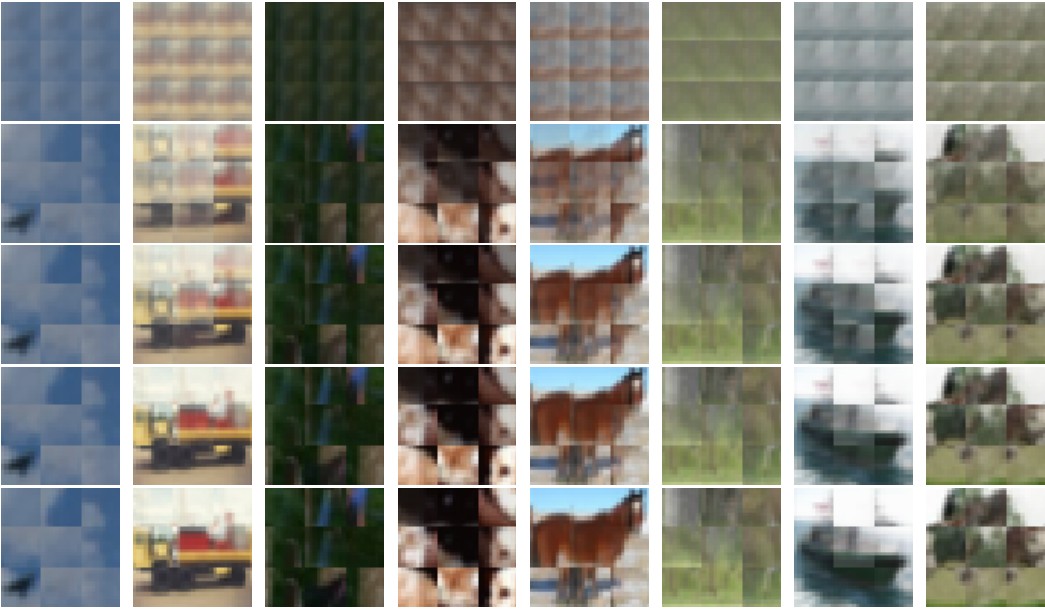

Figure 16: Example reconstructions of PO-U as they are being optimised on CIFAR10 $3 \times 3$ with implicit supervision. These examples have not been cherry-picked.

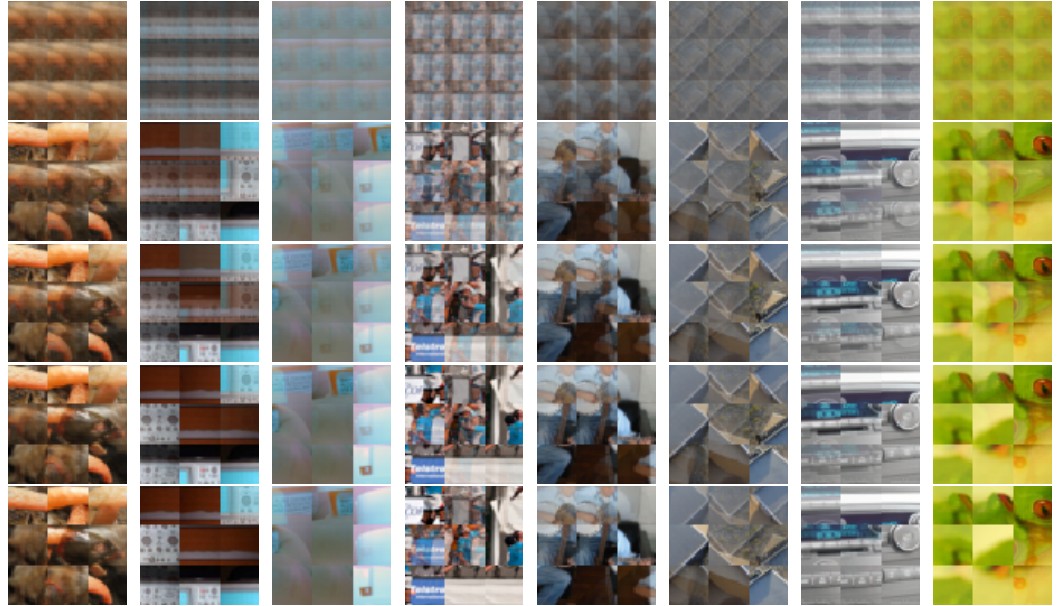

Figure 17: Example reconstructions of PO-U as they are being optimised on ImageNet $3 \times 3$ with implicit supervision. These examples have not been cherry-picked.

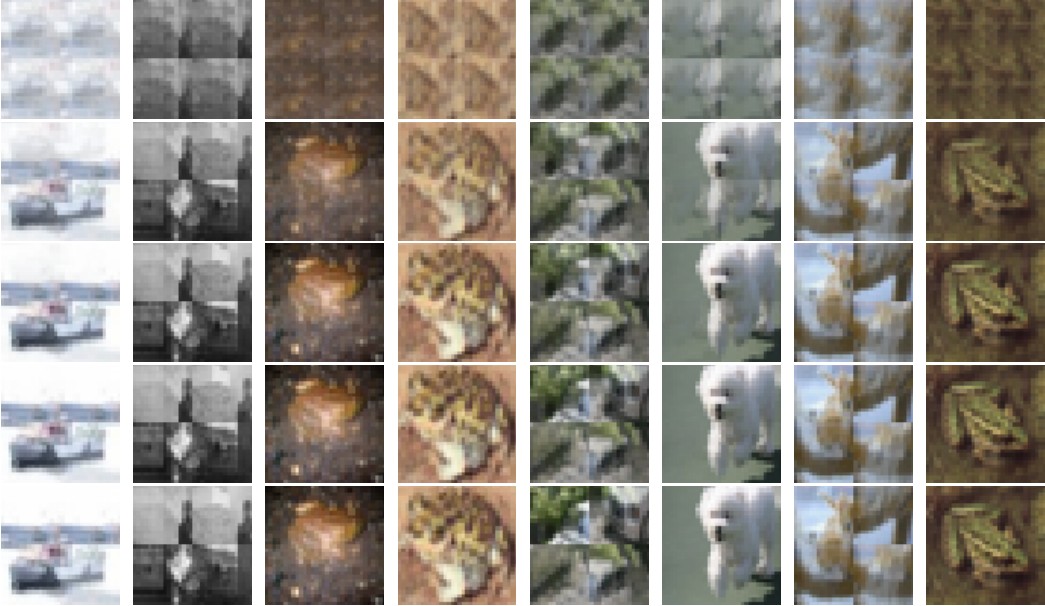

Figure 18: Example reconstructions of PO-U as they are being optimised on CIFAR10 $2 \times 2$ with implicit supervision. These examples have not been cherry-picked.

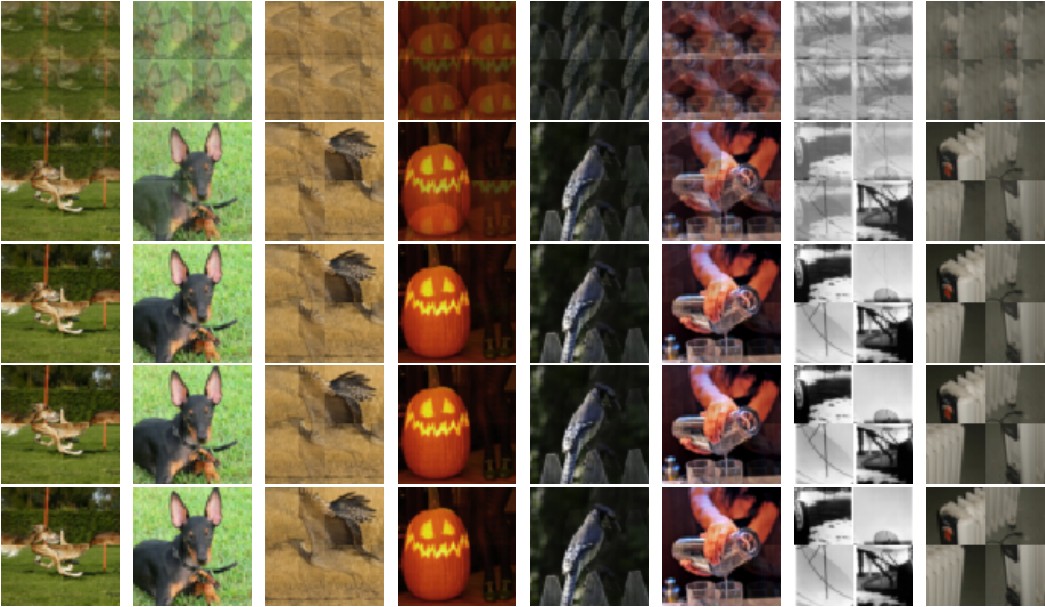

Figure 19: Example reconstructions of PO-U as they are being optimised on ImageNet $2 \times 2$ with implicit supervision. These examples have not been cherry-picked.

