# OpenReview forum: "Learning Representations of Sets through Optimized Permutations"
_ICLR.cc/2019/Conference_

### Official Review · AnonReviewer1 · 2018-10-31
**Interesting idea on learning representations of sets**

**Rating:** 6
**Confidence:** 4

**Review:**

This paper proposed an interesting idea of learning representations of sets by permutation optimizations. Through learning a permutation of the elements of a set, the proposed algorithm can learn a permutation-invariant representation of that set. To deal with the underlying difficult combinatorial optimization problem, the authors proposed to relax the optimization constraints and instead optimize over the set of doubly-stochastic matrices with reparameterization using the Sinkhorn operator. The cost function of this optimization is related to a pairwise ordering cost, which compares the order for each pair of the elements.

The idea of using pairwise comparison information to learn permutations is interesting. The total cost function utilizes the comparison information and optimization over this cost function can lead to a permutation-invariant representation of the set. The idea of using the Sinkhorn operator to reparameterize the doubly-stochastic matrices makes the optimization objective differentiable. Also, the experiment results compared with some baseline algorithms showed the success of the proposed methods in many different tasks.

My major concern of the proposed method is on whether this method can be applied to large sets. Since the algorithm compares all pairs of elements in the set, we need O(N^2) comparisons for a set of size N and hence the proposed method might be slow if N is large. Is it possible to improve the efficiency for large sets?

Questions and Suggestions:

1. Since the authors wants to approximately solve the objective function in Equation (2), it is better if we can see a proof showing why this optimization problem is difficult.

2. For the experiment in Section 4.2, it seems that all methods (including the proposed methods and the baseline methods) are not performing well if the images are split to at least 4 * 4 equal-size tiles. I understand that currently the authors applied their method to the case of grid permutation by simply adding all cost functions of all rows and columns. Is it possible to extend the proposed method to the grid case in another way so that the results under this setting is better?

3. It will be better if the authors can propose some more insights (probably with some theoretical analysis) when can the PO-U method performs better and when can the PO-LA method performs better.

4. The authors mentioned that, the proposed method can get good permutations even for only T=4 steps. What if we continue running the algorithm? Will the permutation converges stably?

5. The authors proposed to update the permutation matrix parameters in an alternative way (Equation (7)) and mentioned that this update works significantly better in the experiments. It will be great if the authors can have a theoretical analysis on why this is true since P and \tilde P can be quite different from each other for an arbitrary \tilde P matrix.


Minor comment:

I think there is a typo in Equation (5). The entry \tilde P_{pq} is related to not only the entry P_{pq}, but also the other entries of the matrix P. Hence, I think Equation (5) should be modified as a matrix multiplication.

---

> ### Author Response · Authors · 2018-11-09
> **Author's response**
>
> Thank you for the review.
>
> Scalability to large sets is indeed an issue with our model. As we discuss in section 5, its intended use is more with small sets of complex elements (e.g. objects in images) rather than large sets of simple elements (e.g. point clouds), with our VQA experiment being an instance of the former.
> For large sets there are some optimisations that can be made. For example, if there are many pairs of elements where we can guess that they will not affect each other's local ordering much (e.g. points in point clouds that are far away from each other), we can make the comparisons  sparse by only comparing points that are reasonably close to each other, which just involves a pre-processing step. We also mention in the Discussion section that a divide-and-conquer strategy (perhaps merge-sort-like) could work, reducing the comparisons down to O(n log n), though it might assume transitivity of the cost function.
>
> 1. Eqn (2), without any relaxations, is a standard Quadratic Assignment Problem (see Appendix C to turn (2) into a more standard formulation). These problems are known to be NP-hard, and even epsilon-approximability is NP-hard [ https://dl.acm.org/citation.cfm?id=321975 ]. For the quadratic programming formulation (when constraints on P are relaxed), we already cite (Pardalos & Vavasis, 1991) in our paper for NP-hardness.
>
> 2. There is a bit of a misunderstanding in interpreting the 4x4 and 5x5 results because we did not explain the metric sufficiently. Especially for MNIST, many of the tiles end up completely blank or very similar when the image is split into many small tiles. This means that there are many equally-valid solutions where blank tiles are assigned to different positions. However, the accuracy metric does not account for these multiple possible solutions properly (only one of these is considered to be the ground truth), so all the blank tiles must happen to be assigned to the blank spots in the ground-truth order for the accuracy to be 1.
>
> We will improve this by swapping accuracy (Table 1) with mean squared error (Appendix E Table 4) for this experiment. Until we have updated the paper, you can see in Table 4 that while there is still a worsening of error as we increase the number of tiles, the change is less abrupt and the models do decently on MNIST for higher number of tiles.
>
> As to ideas for improving results in the setting of many tiles for image mosaics: in the grid case, it is possible to not only have row-wise and column-wise permutations, but also over diagonals, which should help in constraining what permutations are considered good by the model. It is also possible to modify the row-wise comparison to not act on each row individually, but consider assignments to other rows simultaneously (vice versa for columns).
> In some initial work we had some success on image mosaics with an alternative cost function for (1) that only considers correct ordering between direct neighbours, but it struggled even with small instances of the sorting task so we did not pursue this further. It may be possible to fix this issue, perhaps by doing a convex combination of the cost function in the paper and this modified cost function, which should give better results.
>
> 3. As theoretical analysis of these complex learned systems is rather difficult (it is a bit like asking for theoretical analysis of when, for example in the context of sequence modeling, CNNs are better than RNNs), we instead point to the results in section 4: PO-LA is suitable for when LinAssign is decent, which is the case when absolute positioning of set elements is useful (since LinAssign only directly models absolute positioning). The PO part is then used to refine this. PO-U is suitable for when relative position between set elements becomes more important, resulting in LinAssign only learning things that are not useful and thus being a detractor. If the input set can have variable size (e.g. in VQA), it is not possible to use PO-LA, only PO-U.
>
> 4. We took a net that was trained to sort 10 numbers with T=4 and evaluated it with T=100 000. The accuracy degraded from 100% accuracy to ~94% accuracy. We also took a net trained with T=20 and evaluated it with T=100 000, which resulted in 100% accuracy this time. It appears that if the permutations seen in training are sufficiently converged already -- which is the case for T=20 -- it converges stably at evaluation time when run for signficantly longer too.
>
> 5. We will add this as an appendix to the paper. By looking at the gradients before and after the Sinkhorn, it becomes clear that gradients vanish when trying to differentiate through the Sinkhorn the further away from the uniform initialisation one gets. That means that it is difficult to learn Ps that are close to proper permutation matrices (all 0 and 1 entries) if we do not use the alternative gradient.
>
> Typo in equation (5): Good catch! We will fix this in the revision of our paper.

---

> > ### Comment · AnonReviewer1 · 2018-11-14
> > **Some questions**
> >
> > Thanks for the response. Just some small questions:
> >
> > 1. For your response to my first question (on proving the optimization problem is difficult), you can not say this problem is NP-hard by writing it in a Quadratic Assignment Problem (QAP) form and using the fact that QAP is NP-hard to show your problem is NP-hard. Instead, you need to show that, every instance of the QAP problem can be transformed into one instance of your optimization problem efficiently (i.e. the opposite way).
> >
> > 2. As you mentioned in your response to my second question, if we split the images into many tiles, then many of the tiles will be very similar. Is this a problem for only MNIST or for all of the datasets?
> >
> > 3. You mentioned in your response to my fourth question that, the experiment that sorts 10 numbers with T=100000 steps has the accuracy go down from 100% to ~94%. How does this accuracy change during this process? Does the accuracy go up and down during the 100000 steps or the accuracy just goes down once at some point? Will the accuracy go back to 100% if T is larger? Also, what about the other experiments? Will the performances be stable if T is larger than 4?

---

> > > ### Author Response · Authors · 2018-11-14
> > > **Answers**
> > >
> > > 1. You are of course correct and we fully concede that we were sloppy with making those claims. What we meant is that we only managed to find mappings of the problem to known NP-hard problems (which is clearly not a proof that it is NP-hard itself, but perhaps evidence that it is non-trivial). The computational complexity of the optimisation problem is somewhat tangential anyway, since the much more important aspect is the differentiability of the optimisation, which allows the model to be learned end-to-end. This is difficult without the relaxations on P since there would be no meaningful gradient for it. We will fix the one comment in the main text about the complexity of the problem, thank you for pointing this out.
> > >
> > > 2. As we discuss in the paper, this is a problem for the other datasets too. If the number of tiles is the same as the number of pixels, it is obvious that this is an issue with any dataset that can have two of the same pixels anywhere in the image. As we point out in the text, we believe that this manifests itself through the difference of performance between PO-U and LinAssign between CIFAR10 and ImageNet: PO-U performs closer to LinAssign on CIFAR10 where this is a bigger issue (smaller tile sizes, more likely to have similar tiles) than on ImageNet (64x64 images, so bigger tile sizes and less likely to have similar tiles), where PO-U performance is much closer to PO-LA than LinAssign. With VQA this is less likely an issue: since each element of the set is a 2048d vector, unless two elements are describing the exact same object -- in which case this isn't a problem -- it is very unlikely for two elements to be very similar to each other.
> > >
> > > 3. For sorting, accuracy monotonically gets worse than 100% starting from T=10, T=25 gets about 99%, T=50 gets about 0.96%, T=100 gets about 94.5% and beyond T=200 gets 94%. We do not observe any oscillations outside of measurement errors. Thus, we doubt that it could go up again with even larger T considering how smooth this change with increasing T is.
> > > For the image mosaic experiments, it appears to be similarly stable. Again these results should get a fair bit better if the models are trained on higher T too, rather than being evaluated for much longer than it was trained at. We evaluate with T=100 instead of T=100k now since these datasets take longer to run than the simple number sorting. We checked that results don't change much when increasing T over 100 and did not observe significant oscillations.
> > >
> > > Mean squared errors and accuracies, before (eval with T=4 with results from paper) -> after (eval with T=100)
> > > ---------------------
> > > MNIST 2x2: MSE 0.00 -> 0.00, accuracy 100% -> 99.9%
> > > MNIST 3x3: MSE 0.02 -> 0.07, accuracy 65.9% -> 55.6%
> > > MNIST 4x4: MSE 0.46 -> 0.78, accuracy 0.2% -> 0%
> > > MNIST 5x5: MSE 0.45 -> 0.96, accuracy 0% -> 0%
> > > CIFAR10 2x2: MSE 0.11 -> 0.17, accuracy 87.3% -> 85.6%
> > > CIFAR10 3x3: MSE 0.44 -> 0.60, accuracy 34.4% -> 35.0%
> > > CIFAR10 4x4: MSE 1.23 -> 1.39, accuracy 0% -> 0%
> > > CIFAR10 5x5: MSE 1.26 -> 1.48, accuracy 0% -> 0%
> > >
> > > For the classification setting from reconstructed mosaics, the change in classification accuracy is as follows:
> > >
> > > Classification accuracies, before (eval with T=4 with results from paper) -> after (eval with T=100)
> > > ---------------------
> > > MNIST 2x2: 99.4% -> 98.9%
> > > MNIST 3x3: 98.7% -> 92.6%
> > > MNIST 4x4: 67.9% -> 22.9%
> > > MNIST 5x5: 69.2% -> 20.7%
> > > CIFAR10 2x2: 70.8% -> 67.7%
> > > CIFAR10 3x3: 41.6% -> 33.3%
> > > CIFAR10 4x4: 33.3% -> 27.9%
> > > CIFAR10 5x5: 32.3% -> 22.9%
> > >
> > > MNIST on 4x4 does terribly, presumably due to the magnified issues with resolving blank tiles. CIFAR10 does not degrade as much as MNIST with increasing number of tiles.

---

> ### Author Response · Authors · 2018-11-16
> **Paper updated**
>
> In the updated paper, we justify the alternative gradient update (Appendix D) and we fixed the mistakes you noticed (Eqn 5 and comment about hardness). We also explain in the accuracy table of experiment 2 (table now moved to the appendix) why the accuracy of all models is bad for 4x4 tile numbers.

---

### Official Review · AnonReviewer3 · 2018-11-02
**Reinventing the (methodological) wheel?**

**Rating:** 3
**Confidence:** 2

**Review:**

Update: From the perspective of a "broader ML" audience, I cannot recommend acceptance of this paper. The paper does not provide even a clear and concrete problem statement due to which it is difficult for me to appreciate the results. This is the only paper out of all ICLR2019 papers that I have reviewed / read which has such an issue. Of course for the conference, the area chair / program chairs can choose how to weigh the acceptance decisions between interest to the broader ML audience and the audience in the area of the paper.

----------------------------------------------------------------------------------------------------------------------------------

 This paper addresses the problem that often features are obtained as a set, whereas certain orders of these features are known to allow for easier learning. With this motivation the goal of this paper is to learn a permutation of the features. This paper makes the following three main contributions:
1. The idea of using pairwise comparison costs instead of position-based costs
2. The methodological crux of how to go from the pairwise comparison costs to the permutation (that is, solving Eqn. (2) using  Eqn. (1) )
3. An empirical evaluation

I like the idea and the empirical evaluations are promising. However, I have a major concern about the second contribution on the method. There is a massive amount of literature on this very problem and a number of algorithms are proposed in the literature. This literature takes various forms including rank aggregation and most popularly the (weighted) minimum feedback arc set problem.  The submitted paper is oblivious to this enormous literature both in the related work section as well as the empirical evaluations. I have listed below a few papers pertaining to various versions of the problem (this list is by no means exhaustive). With this issue, I cannot give a positive evaluation of this submitted paper since it is not clear whether the paper is just re-solving a solved problem. That said, I am happy to reconsider if the related work and the empirical evaluations are augmented with comparisons to the past literature on the methodological crux of the submitted paper (e.g., why off-the-shelf use of previously proposed algorithms may or may not suffice here.)


Unweighted feedback arc set:

A fast and effective heuristic for the feedback arc set problem, Eades et al.

Efficient Computation of Feedback Arc Set at Web-Scale, Simpson et al.

How to rank with few errors, Kenyon-Mathieu et al.

Aggregating Inconsistent Information: Ranking and Clustering, Ailon et al.


Hardness results:

The Minimum Feedback Arc Set Problem is NP-hard for Tournaments, Charbit et al.


Weighted feedback arc set:

A branch-and-bound algorithm to solve the linear ordering problem for weighted tournaments, Charon et al.

Exact and heuristic algorithms for the weighted feedback arc set problem: A special case of the skew‐symmetric quadratic assignment problem, Flood

Approximating Minimum Feedback Sets and Multicuts in Directed Graphs, Even et al.


Random inputs:

Noisy sorting without resampling, Braverman et al.

Stochastically transitive models for pairwise comparisons: Statistical and computational issues, Shah et al.

On estimation in tournaments and graphs under monotonicity constraints, Chatterjee et al.


Survey (slightly dated):

An updated survey on the linear ordering problem for weighted or unweighted tournaments, Charon et al.


Convex relaxation of permutation matrices:

On convex relaxation of graph isomorphism, Afalo et al.

Facets of the linear ordering polytope, Grotschel

---

> ### Author Response · Authors · 2018-11-07
> **Previous methods cannot be used for learning**
>
> Thank you for the review. The key difference you missed that separates our work from the papers you cite is that our method is differentiable. In our problem set-up, we are not given the pairwise costs; they have to be learned. In order for these costs to be learnable with gradient descent, we have to be able to differentiate through the algorithm. This is possible with our method, but not possible with traditional literature on feedback arc sets. Experimental comparisons to the papers you list are thus not meaningful, since the costs that these algorithm operate on have to be learned first. Does this sufficiently clarify for you why our methodology is not reinventing the wheel?
>
> We already cite the particular convex relaxation of permutations that we use (Fogel et al., 2013; Adams & Zemel, 2011) and the NP-hardness of the problem (Pardalos & Vavasis, 1991).
>
> Though we mention this matter of differentiability several times throughout the paper, we will add a sentence in the Related Works section to make this distinction with the work on feedback arc sets even clearer.

---

> > ### Comment · AnonReviewer3 · 2018-11-08
> > **Isn't the cost function computed separately?**
> >
> > Okay so then it is not clear to me what the paper is doing. My interpretation of the writing is that the cost function is computed __separately__ from optimising the permutation. For example the beginning of Section 2.2 says "Now that we can compute the total cost of a permutation, we want to optimise this cost with respect to a permutation." Is that not true? If so, then could you please provide a complete and self-contained problem statement? Thank you.

---

> > > ### Author Response · Authors · 2018-11-08
> > > **Separate in forward pass, but not when doing backpropagation to learn it**
> > >
> > > It is indeed computed separately in the forward pass of the model. But during training, backpropagation adjusts the weights that the cost function depends on, based on the gradients that are backpropagated through the algorithm.
> > >
> > > We have a model that is given some input data, where each individual data sample is a set, and some target output. During the forward pass (first half of one training step, or one full inference step), a neural net turns the set into the matrix of pairwise costs and our permutation algorithm is run on it to produce a permutation matrix. The permutation matrix is applied to the set to produce a sequence, which can now be processed further by another neural net to produce the predicted output. During the second half of this training step, gradients of a loss function are backpropagated through the network to minimise how different the predicted and target outputs are. To backpropagate through to the weights that determine what the cost function is, we have to backprop through the algorithm first. At the start of training a network, all weights are random so the permutation produced is nonsense, since the neural net that determines the cost function has not learned what permutations are good for the particular task is yet. Through training, the neural net that computes the cost function ($f$ in the paper) receives gradients (by backpropagating the gradients of the loss function through the permutation algorithm) to learn how to assign costs to inputs appropriately.

---

> > > > ### Comment · AnonReviewer3 · 2018-11-12
> > > > **Please provide a formal problem statement**
> > > >
> > > > Thank you for your response. As I requested in my  previous comment, "could you please provide a complete and self-contained problem statement?"  Please provide a mathematically rigorous __problem__ statement, and not the proposed method of solving it. Thank you.

---

> > > > > ### Author Response · Authors · 2018-11-12
> > > > > **Problem statement**
> > > > >
> > > > > We are given an input that is a set of cardinality $N$ of feature vectors with dimensionality $M$, represented as a matrix $X \in \R^{N \times M}$ with the feature vectors as rows in some arbitrary order. The problem is to find a function $g: R^{N \times M} \to R^{N \times M}$ that permutes the rows of $X$ into a sequence of feature vectors $g(X) = Y$, with the constraints that $h$ must be:
> > > > >
> > > > > 1. invariant to permutation of rows of X, i.e. $g(X) = g(PX)$ for all permutation matrices $P$,
> > > > > 2. differentiable almost everywhere.
> > > > >
> > > > >
> > > > > This is essentially the problem statement that we give in the first paragraph of section 2. Note that we are doing something much more general compared to the papers you cited in your review:
> > > > > 1. there is no ground-truth of what the cost/weight between set elements is,
> > > > > 2. there is no ground-truth notion of what a "good" output sequence $Y$ is.
> > > > >
> > > > > These aspects can be learned in the usual Empirical Risk Minimisation setting. The differentiability constraint is necessary as we are interested in using deep neural networks trained with gradient descent to provide the input set $X$ and process $h(X)$ further.

---

> > > > > > ### Comment · AnonReviewer3 · 2018-11-19
> > > > > > **Again, please provide a formal problem statement**
> > > > > >
> > > > > > In your comment, what is g and what is h? It says the goal is to design g but the comment uses notation h twice, where h is undefined.
> > > > > >
> > > > > > Moreover, is g restricted to be a permutation ("problem is to find a function $g: R^{N \times M} \to R^{N \times M}$ that permutes")?
> > > > > >
> > > > > > Can you please provide a proper formal proper problem statement?

---

> > > > > > > ### Author Response · Authors · 2018-11-20
> > > > > > > **What are you looking for precisely?**
> > > > > > >
> > > > > > > What exactly do you understand as a proper formal problem statement? What is insufficient about the problem statement that we gave?
> > > > > > >
> > > > > > > We meant $g$ where we used $h$, it was a typo.
> > > > > > >
> > > > > > > Differentiability almost everywhere and permutation-invariance are both formally well-defined properties of functions. We did not specify what "permutes X" means exactly to not tie it to our method of matrix-multiplying PX, where P is a doubly-stochastic matrix in our case. This was in response to your previous complaint of tying down the description with our specific method of solving it.

---

> > > > > > > > ### Comment · AnonReviewer3 · 2018-11-20
> > > > > > > > **What I am looking for precisely**
> > > > > > > >
> > > > > > > > I am looking for a proper problem statement which still has not been specified.
> > > > > > > >
> > > > > > > > Thanks for clarifying the g vs h errors in your previous comment.
> > > > > > > >
> > > > > > > > If the only requirements are
> > > > > > > > "1. invariant to permutation of rows of X, i.e. $g(X) = g(PX)$ for all permutation matrices $P$,
> > > > > > > > 2. differentiable almost everywhere."
> > > > > > > > then setting g as a constant function satisfies both constraints. So there is more to the problem than just these two requirements. I am looking for precisely what this problem entails.

---

> > > > > > > > > ### Author Response · Authors · 2018-11-20
> > > > > > > > > **Empirical Risk Minimisation**
> > > > > > > > >
> > > > > > > > > We believe that you are looking for the Empirical Risk Minimisation [0] or Structural Risk Minimisation framework that we mentioned in one of our earlier comments. Sections 1 to 4, 7, and 8 in that seminal paper should hopefully clear up the formal setting of ERM and SRM for you. This is the usual setting for supervised learning problems, which includes what we consider in our paper.
> > > > > > > > >
> > > > > > > > > Through a priori knowledge of the problem (learning representations of sets), we add a structure with specific properties (differentiability, permutation invariance) into the neural network.
> > > > > > > > >
> > > > > > > > >
> > > > > > > > > [0]: V. Vapnik. Principles of Risk Minimization for Learning Theory. In NIPS, 1992. http://papers.nips.cc/paper/506-principles-of-risk-minimization-for-learning-theory.pdf
> > > > > > > > >
> > > > > > > > > Edit: Apologies if this comment perhaps seemed a bit condescending, but we do not know your background and without you being more precise about what your imagined problem statement contains, we do not know what to answer.

---

> > > > > > > > > > ### Comment · AnonReviewer3 · 2018-11-20
> > > > > > > > > > **-**
> > > > > > > > > >
> > > > > > > > > > I'm sorry that there seems to be a communication gap. I am well aware of ERM/SRM etc. However I don't see your problem description specifying what training data is available etc. for (supervised) learning. That is why I asked for a formal problem statement but the one provided seems to be incomplete.
> > > > > > > > > >
> > > > > > > > > > Anyways, this is my last comment since I have spent a highly disproportionate amount of time in trying to get a formal problem statement. I think this paper may be suitable for people who are in this specific ballpark of area of research and I will leave it up to the other reviewers, but for someone like me who is somewhat farther away, I feel it is unfortunate that the problem statement is also not specified clearly. If the results are indeed good (and I hope they are) then it would be very useful for the authors if the paper was made more accessible to a broader ML audience.

---

> > > > > > > > > > > ### Author Response · Authors · 2018-11-21
> > > > > > > > > > > **Revised problem statement with complete context**
> > > > > > > > > > >
> > > > > > > > > > > We recognise, and thank, you for the time that you have spent on this review. Looking back through the discussion, we believe we might now have a clearer idea where you are having difficulty with understanding our approach. Our formal problem statement above focuses on the specific contribution of the paper: proposing a new module that can be inserted within a deep network to explicitly allow for permutation invariance of the inputs whilst producing a single fixed output list. Such an approach is crucial for any learning task where we believe that set representations (or equivalently permutation invariance) is important (as demonstrated through the many experiments in the paper).
> > > > > > > > > > >
> > > > > > > > > > >
> > > > > > > > > > > We will end with a revised formal problem statement that covers the entire problem context. $X$ and $f$ used here is unrelated to the $f$ in the paper. $g$ matches the $g$ in our problem statement above.
> > > > > > > > > > >
> > > > > > > > > > > Assume that we have some learning task that involves learning a complex mapping between inputs X and outputs y. We choose to solve this problem with a deep network formulated as two parts: a deep feature extractor $f(X, \theta_f)$ that produces a set of feature vectors and a classifier $h(f(X, \theta_f), \theta_h)$ that produces $\hat{y}$, which are estimates of $y$. $\theta_f$ and $\theta_h$ are the parameters of the neural networks $f$ and $h$ respectively. We emphasise that the outputs of $f$ should be treated as a mathematical set, even though they would be encoded as a list of feature vectors.
> > > > > > > > > > >
> > > > > > > > > > > $h$ should properly treat its input as a set, but it is difficult to structure and learn the parameters of $h$ in such a way that the outputs of it will be the same for any permutation of the feature vector list produced by $f$. Our proposed change to the problem is to add a learnable module -- $g(\cdot, \theta_g)$ -- between $f$ and $h$ that transforms the list of feature vectors that $f$ produces into a canonical representation that is the same for any permutation of that list.
> > > > > > > > > > >
> > > > > > > > > > > Our complete learning problem can thus be expressed as one of finding parameters $\theta_f, \theta_g, \theta_h$ such that the empirical risk with a given loss function between true $y$s and estimated $\hat{y}$s, where $\hat{y} = h(g(f(X, \theta_f), \theta_g), \theta_h)$, is minimised.
> > > > > > > > > > >
> > > > > > > > > > > To achieve this, we use gradient descent to train the parameters $\theta_f, \theta_g, \theta_h$. This requires that $f, g, h$ are differentiable. To satisfy the requirement of $g$ producing a canonical representation regardless of the input ordering, $g(f(X, \theta_f), \theta_g)$ must equal $g(P f(X, \theta_f), \theta_g)$ where P is any permutation matrix and the list of feature vectors (which is to be treated as a set) that f produces are placed in the rows of a matrix.

---

> ### Author Response · Authors · 2018-11-16
> **Have your concerns been addressed?**
>
> Have we explained why our methodology is not just reinventing the wheel sufficiently? We have updated the related works section with a citation to the survey you referenced and explain why that line of work is not relevant to what we are doing.

---

### Official Review · AnonReviewer2 · 2018-11-04
**An interesting method to learn (latent) permutations based on pairwise costs.**

**Rating:** 6
**Confidence:** 4

**Review:**

The authors introduce a method to learn to permute sets end-to-end. They define the cost of a permutation as the sum of pairwise costs induced by the permutation, where the pairwise costs are learned. Permutations are made differentiable by relaxing them to doubly stochastic matrices which are approximated with the Sinkhorn operator. In the forward pass of the algorithm, a good permutation (ie one with low cost) is obtained with a few steps of gradient descent (the forward pass itself contains an optimization procedure). This permutation is then either used directly as the output of the algorithm or is used to permute the original inputs and feed the permuted sequence to another module (such as an RNN or a CNN). The method can easily be adapted to other structures such as lattices by considering row-wise and column-wise pairwise relations.

The proposed method is benchmarked on 4 tasks:
1. Sorting numbers, where they obtain very strong generalization results.
2. Re-assembling image mosaics, on which they obtain encouraging results.
3. Image classification through image mosaics.
4. Visual Question Answering where the permuted inputs are fed to an LSTM  whose final latent state is fed back into the baseline model (a bilinear attention network). Doing so improves over feeding the inputs to an LSTM without learning the order.for which the output is the permutation itself and  classification from image mosaics and visual question answering which require to learn an implicit permutation.

The method is most similar to Learning Latent Permutations with Gumbel-Sinkhorn Networks (Mena et al) but considers pairwise relations when producing the permutation. This can have important advantages (such as taking local relations into account, as shown by the strong sorting results) but also drawbacks (inability to differentiate inputs with similar content), but in any case this represents a good step towards exploring with different cost functions.

The method can be quite unpractical (cubic time complexity in set cardinality, optimization in forward pass, having to preprocess the set into a sequence for another module can be resource expensive).
Experimental results on toy tasks (tasks 1, 2 and 3) are encouraging. The approach improves over a relatively strong baseline (task 4) although it isn't clear that it would still hold true when controlling for number of parameters and compute.

I have a few comments about the presentation (for which I would be willing to change my score to a 6):
- When possible, please use the numbers reported by Mena et al and consider reporting error (instead of accuracy) as they do to ease comparison. The results that you report using their method are quite worse than what they report, so I think it would be fair to include both your reimplementation and the initial results in the table.
- It would be interested to have some insights on what function f is learned (for the sorting task and re-assembling image mosaics for example).
- Clarity would be improved with figures representing which neural networks are used at what part of the process.


###########################################
Updated review:

The authors have greatly improved presentation and have addressed concerns about the increase in parameters and computation time. I have changed my score to a 6.

---

> ### Author Response · Authors · 2018-11-09
> **Author's Response**
>
> Thank you for the review.
>
> In general, we intend our model to be useful for relatively small sets of complex objects like in VQA, rather than large sets of simple objects where the cubic time complexity indeed becomes a big problem.
> About your concern whether the results hold up when controlling for parameters and computation: the time increase by using our model compared to the baseline is about 30% (4400 seconds per epoch instead of 3400 seconds). This is a similar increase in computation time as the change from BAN-8 to BAN-12 (increasing number of attention glimpses to 12) in their paper (Kim et al., 2018), Table 1. Their difference is within one standard deviation (0.04% increase, stdev of 0.11%), so simply increasing the BAN model size alone is clearly running into diminishing returns already. Our model does not add a significant number of parameters compared to changing BAN-8 to BAN-12. Our model also results in qualitatively different improvements: general improvements in VQA typically result in a roughly even improvements in all the categories, whereas our model improves on number questions significantly more than other categories. We will make this clearer in the revision.
>
> - We will swap the table of errors (Appendix E Table 4) with the table of accuracies (Table 1) in the main body. We initially decided on reporting accuracy in the main body because MSE may not be directly comparable: we do not know whether their pixel values were scaled to have unit variance (our choice) or to be between 0--1, the latter of which would make their errors seem lower than ours for the same reconstruction (in personal communication, they said that they did not normalise the data, but our results suggest that they did, since our reproduced MSEs on MNIST closely match theirs). As Reviewer 1 seemed to have a slight misunderstanding with accuracy too, we agree that comparing MSE in the main body is likely clearer.
>
> Contrary to what you say, our reproductions on MNIST are fairly close to what they report (our reproduced MSEs are roughly even, accuracies are slightly worse) and only on ImageNet are our reproduced results of their model worse. To be clear, the relevant row in their results to compare our accuracy to is "Prop. any wrong" (reconstruction is correct only if all tiles are correct, this is what we use), not "Prop. wrong" (reconstructions of the tiles being individually correct). As per your suggestion, we will include their results in our tables to make the appropriate comparison easier.
>
> - We will perform some analysis of the learned f and include it as an appendix. For number sorting, it is enough for it to learn f(x_i, x_j) = x_i, so F(x_i, x_j) = x_i - x_j, which is a sensible comparison function. In initial analysis it appears to learn a scaled and shifted version of that. We are currently looking into what it learns for image mosaics.
>
> - We will add some figures to make the network architectures for the different tasks clearer.

---

> > ### Comment · AnonReviewer2 · 2018-11-09
> > **Looking forward to see these encouraging revisions**
> >
> > Just to clarify about the error, I meant (1-accuracy), not MSE and I agree MSE could be harder to compare.
> > Your reimplementation wasn't significantly worse but if I recall correctly there were a few instances for which the following was true: Mena et al > Yours > Your reimplementation of Mena.

---

> > > ### Author Response · Authors · 2018-11-12
> > > **Accuracy versus error**
> > >
> > > Once we update the paper with Mena's results added into our tables, comparison of the results should be easy regardless of whether we report accuracy or (1-accuracy). Unless you feel strongly about this, we prefer to use accuracy, precisely due to this possible confusion between (1-accuracy) error and mean squared error.
> > > The instance you mention where Mena et al > ours > our reimplementation is wrt MSE only on ImageNet 3x3 and wrt accuracy on ImageNet 3x3 and MNIST 4x4. This will become clearer with the update.
> > >
> > > A caveat about their results to be aware of, which we mention in Appendix E.2, is that they upscaled their MNIST images by a factor of 2. When we tried this, it lead to better results for all models (with ordering between models preserved) in our testing too. However, this seemed too arbitrary to us and we decided to not do this. We will mention this detail about their results in the main body because we include their results now.

---

> ### Author Response · Authors · 2018-11-16
> **Paper updated**
>
> We addressed 2 of your 3 comments with our updated paper and will update the paper again soon with the network architecture figures. Let us know if there are other types of qualitative analysis that might be of interest.

---

### Author Response · Authors · 2018-11-16
**Paper revision 1**

We have updated the paper with the following changes:

- Scalability was one of the major concerns of Reviewer 1 and 2. We now specify in section 5 that for the experiments on the real-world dataset VQA using object proposals, this is only an increase of 30% in computation time with set sizes up to 100. It's also useful to keep in mind that the cubic comes from multiplying two NxN matrices together, which is an operation that GPUs are very fast at. Mena et al's simple linear model is O(N^2), so our O(N^3) that properly models pairwise comparisons is not too far off from this.
To put this into perspective, we can compare our O(N^3) to the processing of a length N sequence with an RNN that has a hidden state size of H. The N hidden state updates, which is a matrix-vector product each, has time complexity O(N * H^2). In comparison to our O(N^3), if H > N then our complexity is better, and N > 100 with H > 1000 is not uncommon in tasks such as language modeling.

- We added Appendix C wherein we analyse the learned comparison functions qualitatively (Reviewer 2). This contains some figures that help understand what it has learned, which may be of interest to everyone.

- We added Appendix D wherein we justify using the alternative gradient update in the internal optimisation. While this is not a proof of, for example, convergence speed, this should be enough to give a good idea of why it is useful in our case. (Reviewer 1)

- For experiment 2, we swapped the table of accuracy (previously in main body) with the table of mean squared errors (previously in appendix) because the accuracy metric has some flaws that can lead to misunderstandings. (Reviewer 1)
- We fixed the typo in Eqn (5) (Reviewer 1).
- We fixed the inaccurate statement about computational complexity of the problem (Reviewer 1).
- We now clarify that the benefit on VQA of including our model holds when controlling for computation time and model size in section 4.4. (Reviewer 2)
- For experiment 2, we now include Mena et al's results directly in our tables for ease of comparison, with the caveat that we knowingly avoid an improvement to results that they get on MNIST through arbitrarily upscaling the images. (Reviewer 2)
- We now clarify why existing literature on minimum feedback arc sets is not relevant in section 3. (Reviewer 3)


We are now working on the figures of the network architectures used for the different tasks and hope to update the paper with them very soon. (Reviewer 2)

We thank the reviewers again for the good comments and suggestions, which certainly helped us in improving the paper.

---

> ### Comment · AnonReviewer2 · 2018-11-19
> **Responses to revisions**
>
> The new sections in the appendix provide interesting insights about what is learned and the revisions in the presentation make it easier to compare results with Mena et al.
>
> My concern about the practicality of the approach still remains, although the computation time increase of 30% is reasonable. I'm also curious about which hardware was used and most importantly the increase in number of parameters.

---

> > ### Author Response · Authors · 2018-11-20
> > **Hardware and parameter counts**
> >
> > All the experiments were run on GTX 1080 Ti GPUs. For the sorting experiment and the experiments on MNIST and CIFAR10, we used 1 GPU. For the ImageNet versions, we used 2 GPUs. For VQA, we used 4 GPUs (BAN authors used 4 Titan XP GPUs).
> >
> > Parameter counts:
> > BAN baseline: 85,968,934
> > BAN with PO-U and LSTM: 86,062,128

---

### Author Response · Authors · 2018-11-20
**Paper revision 2**

- We realised that we forgot to mention that the 30% increase in computation time comes not from processing with just 1 PO-U module, but 8 PO-U modules since there are 8 attention glimpses in the BAN model and each is processed separately. The paper now includes this information. The individual modules are thus faster than we initially suggested.
- Figures of network architectures for the 4 different tasks are now included (Figure 9, 10, 11). (Reviewer 2)
- Small wording change in section 2 paragraph 1 to more explicitly state that differentiability is necessary. (Reviewer 3)
- In the title, we changed the spelling from "Optimised" to "Optimized".

We have also updated https://github.com/iclr2019-anon123456/perm-optim with our code for the VQA experiment (all experiments should be reproducible now) and the code for the visualisations that we added in the last revision.

---

### Author Response · Authors · 2018-11-22
**Paper revision 3**

- We have added a reference to the concurrent ICLR submission "Janossy Pooling" https://openreview.net/forum?id=BJluy2RcFm
This work can be considered complementary to ours: they focus on averaging higher-order interactions between set elements (which generalises Zaheer et al.'s and Santoro et al.'s models we already mentioned in our paper) and simply training with randomly sampled permutations, while our work focuses on how to flexibly learn a canonical ordering that the RNN processing the permuted set "likes", rather than giving it a random ordering each time.

---

### Meta-Review · Area_Chair1 · 2018-12-14

**Confidence:** 4
**Recommendation:** Accept (Poster)

**Metareview:**

The paper proposes an architecture to learn over sets, by proposing a way
to have permutations differentiable end-to-end, hence learnable by gradient
descent. Reviewers pointed out to the computational limitation (quadratic in
the size of the set just to consider pairwise interactions, and cubic overall).
One reviewer (with low confidence) though the approach was not novel but
didn't appreciate the integration of learning-to-permute with a differentiable
setting, so I decided to down-weight their score. Overall, I found the paper
borderline but would propose to accept it if possible.